# Electric transmission value and its drivers in United States power markets

Julie Mulvaney Kemp ⬡ ✉, Dev Millstein ⬡, Will Gorman ⬡, Seongeun Jeong ⬡ & Ryan Wiser ⬡

Electric transmission infrastructure plays a vital role during extreme weather and supply disruptions and can enable low-cost electricity systems. This paper contributes to a more complete understanding of the value and cost-effectiveness of transmission, as well as barriers to its development. By studying wholesale energy market prices in the United States between 2012 and 2022, we find that additional transfer capacity between regions would have been especially valuable, with a median value of $116 million per GW per year. This capacity would often have provided balanced benefits to each region. The market value of transmission was highly influenced by a small fraction of time: 5% of hours typically captured at least 45% of the total value. These peak periods were primarily driven by unforeseen changes in conditions within one day of operations. Annualized transmission infrastructure cost estimates were lower than the average market value for most locations, including all links crossing regional seams, where the value-to-cost ratio was often greater than 4. This suggests that there are barriers to developing valuable grid infrastructure. These results complement forward-looking modeling studies and support efforts to improve modeling practices.

Recent events have put electric transmission in the spotlight. In the United States, extreme winter storms have caused electric system outages, leading to discussion of transmission as a possible solution[1,2]. In Europe's 2021–2022 energy crisis, existing well-connected electricity networks increased supply security[3], yet the value of more cross-border transmission capacity was also apparent[4]. Looking beyond individual events, discussion around transmission development has also been elevated by industry trends during the 2010s and early 2020s, including generation fleets shifting away from coal and nuclear toward gas, wind, and solar, record infrastructure investments by China's State Grid[5], and continued siting challenges such as local opposition[6,7] and complex permitting[8]. In the US, a slowdown in construction of new high-voltage lines[9] suggests there may be pent-up demand for transmission, while transmission also serves an increasing role in accessing generation resources in new locations. Transmission is also considered a key enabler of a low-cost and low-carbon electricity system[10–13]. Motivated by this confluence of factors, improving

transmission planning processes and expanding transmission is a focus of regulators[14], legislators[3,15–17], and private sector analyses[4,18,19].

Transmission development in the United States typically occurs through one of three channels: centrally planned through a regional system operator (most common), independently constructed by a merchant transmission developer (rare), or generator interconnection processes (frequent, mostly incremental upgrades). In the latter channel, development is narrow in scope, including only those upgrades required to maintain grid safety, reliability, and often deliverability after adding specific new resources. In the first two channels, development decisions come down to a cost/benefit trade-off, though system operators and merchants may assess different sets of costs and benefits. Traditionally in centralized planning processes a need for transmission is established to either ensure reliability or satisfy public policy directives. Economically driven transmission projects have also been pursued and are typically evaluated on their ability to reduce production costs of the system. However, modeled production cost

Lawrence Berkeley National Laboratory, Berkeley, CA, USA. ✉e-mail: jmulvaneykemp@lbl.gov

benefits alone rarely outweigh transmission costs[20]. In recognition of the many ways transmission can be beneficial, including lower generation capital costs, reduced pollution, risk mitigation, and reducing unserved demand, in addition to traditional production cost savings[21], some system planners have moved to a multi-value study framework[22].

To account for system changes expected during a transmission asset's lifetime, cost/benefit analyses are typically performed by simulating the electricity system under assumptions about future system conditions. A weakness of such models is that often they do not replicate real-world conditions that affect the benefits of transmission, conditions such as forecast errors, extreme weather, and infrastructure outages[18,19,23,24]. Because transmission infrastructure takes many years to plan and build and, by nature, affects multiple locations, the impacts of misestimating transmission value during the planning phase could be long lasting and widespread.

The purpose of this paper is to provide insight into transmission value by offering a direct contrast to forward-looking modeling. We use observed wholesale market prices, along with load and renewable data, to assess where, when, and why transmission is valuable, focusing on 70 location pairs across the contiguous US from 2012 through 2022. We also conduct a scoping-level analysis to compare historical energy market values of transmission to the costs of transmission infrastructure projects while accounting for market depth, i.e., the extent to which prices would change as a result of new capacity. These comparisons contextualize the empirical value estimates and highlight where market barriers including, but not limited to, current planning processes are greatest.

Specifically, we analyze locational marginal prices (LMPs) that measure the marginal cost of serving the next increment of demand at a specified pricing node and reflect the sum of three components: system-wide marginal energy cost, marginal cost of losses, and marginal cost of congestion. LMPs are defined by the seven independent system operators (ISOs) and regional transmission organizations (RTOs) in the US (i.e., CAISO, ERCOT, ISO-NE, MISO, PJM, and SPP) at over 80,000 nodes, typically for both a forward day-ahead market and a spot real-time market. Concurrent price differences between market nodes indicate network congestion and reflect challenges due to actual generator and infrastructure outages. Market prices serve as an investment signal to market actors and are a rich data source that have been used by researchers to advance our understanding of the value of renewable energy[25,26], efficient levels of subsides for energy efficiency[27], and the impact of specific transmission investments that are already online[28–30], for example.

While our approach is not meant to replace models, it is intended to help identify key mechanisms that may lead to biased model estimates of transmission value. Future efforts could use these insights to improve transmission planning processes. There are several categories of societal benefits that transmission may provide, including resource adequacy, resilience, risk mitigation, and reductions in emissions, market power, capital costs and production costs[21]. These concepts are priced into wholesale electricity markets to varying degrees, and similarly reflected in market values of transmission. Centralized system planners, however, separately quantify each benefit they consider and typically use production cost savings as the main economic benefit. We do not offer a one-to-one comparison between transmission's market value and each of its modeled benefit types, instead using market value as an aggregate signal. Empirical prices precisely reflect actual system conditions and market participant behavior as they occurred in the past, providing insights that cannot be gained from system models.

Prior to this work, studies of transmission value using historical market price data focused on select severe weather[1,2] or geopolitical[4] events, the specific benefits of increased competition[31,32], measuring the impact of a specific transmission investment that is already online[28–30], or congestion on existing lines within a limited footprint (ISO/RTO annual reports, such as[33–35]). The authors of this paper have

previously employed a subset of the methods used here and published the findings in refs. [36–38]. This paper contributes to a more complete understanding of transmission's value, cost-effectiveness, and market barriers and aids efforts to improve modeling practices by identifying key patterns and value drivers for use in model validation.

## Results

### Geospatial patterns of transmission value

The transmission value analyzed in this section is a pairwise quantity between two wholesale market pricing nodes (i.e., a "link") defined as the mean absolute locational marginal price difference between the nodes over time (in units $/MWh). It represents the marginal energy value of spatial price arbitrage within a specific market interval. A link is uniquely defined by two pricing nodes; it does not correspond to a specific transmission line, and there may be zero, one, or multiple existing transmission paths between a link's nodes. This paper primarily focuses on the real-time market, because we are most interested in transmission's ultimate value to the system and real-time prices reflect physically binding dispatch decisions under the actual operating conditions of the system. Further, only real-time prices exist for all 70 links depicted in Fig. 1, though day-ahead prices, derived from a forward financial market, exist for many of the links. We will later offer a comparison to value based on prices from the day-ahead timeframe. The marginal value is useful as it can be directly compared to market prices, but it can also be converted to an equivalent total value per link, such as millions $/GW. Total value metrics are useful when compared to transmission costs. When considering total value metrics, it is important to also consider market saturation effects (e.g., in each hour, is the full GW of transmission capacity needed, or would prices converge with less capacity?).

Seventy hypothetical transmission links between major market regions in the contiguous U.S. are considered; 30 links are contained within a balancing authority ("within-region"), 31 links are interregional within the same interconnection ("interregional") and 9 links span an interconnection seam ("cross-interconnect"). In the contiguous U.S. there are three interconnections – the Western Interconnection, the Eastern Interconnection, and the Electric Reliability Council of Texas (ERCOT) – that operate almost independently with separate frequency synchronization and limited cross-interconnect transfer capacity. Each line segment in Fig. 1a and each point in Fig. 1b show the marginal transmission market value for an individual link, averaged over the study period. Relatively high value links are found within and between many regions.

Generally, links crossing market seams have greater potential to arbitrage high and low prices: The market value for the median cross-interconnect and interregional links are $30/MWh and $15/MWh, respectively, compared to $8/MWh for a within-region link. For context, if the marginal value of $15/MWh was maintained for the entire capacity of a 1 GW line, the line's market value would be $131 million per year $\left(\frac{\$15}{\text{MWh}} * \frac{1000\text{MW}}{1\text{GW}} * \frac{8760\text{h}}{\text{year}}\right)$. Note that market saturation effects will be quantified and discussed later. Interregional and cross-interconnect links may have higher value due to more diversity of weather, load profiles, and generator resources than is found within regions and due to a historical focus by transmission system operators on development within their own geographic footprint[39].

Links connecting ERCOT to any of its neighbors (SPP, MISO, non-ISO West) register the highest values, driven by exceptionally high values since 2018. Links bridging the western and eastern interconnections (i.e., between the West and SPP or MISO) and connecting NYISO and ISO-NE are also among the most valuable. ERCOT, SPP, and NYISO contain the highest value within-region links. Not all links have substantial value, however. For example, the average hourly price difference between ISO-NE's Massachusetts and Maine hubs is just $2/MWh, consistent with ISO-NE's statement that "transmission system

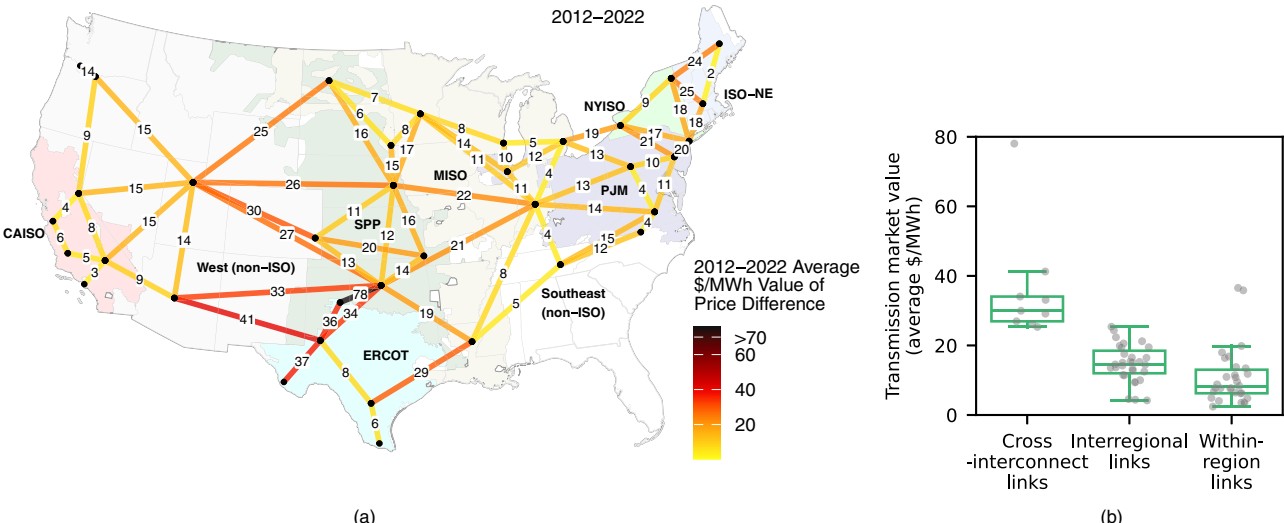

(a)

(b)

**Fig. 1 | Geospatial patterns of transmission value. a** Map of mean marginal transmission market values for all 70 analyzed links over the entire study period (real-time market). The line segments depict which pairs of wholesale market pricing nodes are analyzed and do not portray existing transmission lines. **b** Distribution of mean marginal transmission market values (real-time market) across the set of 70 analyzed links. Each point represents one link, and there are the following number of links in each category: cross-interconnect: 9, interregional: 31, within-region: 30. The horizontal lines on each box plot show, from low to high, the smallest data point lying within 1.5x the inter-quartile range (IQR) from the 25th percentile, the 25th percentile, the 50th percentile (median), the 75th percentile, and the largest data point lying within 1.5x the IQR from the 75th percentile.

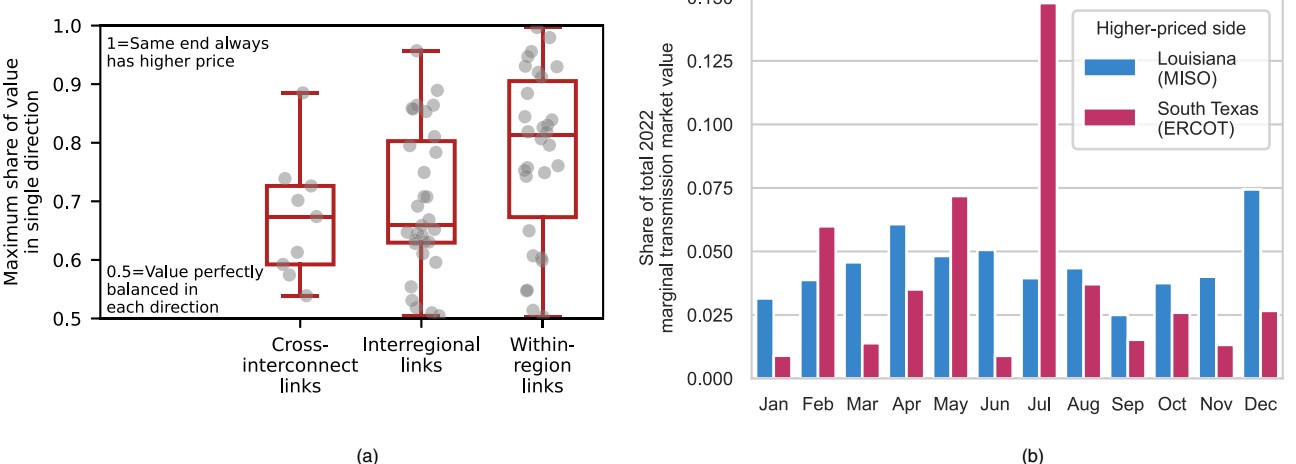

(a)

(b)

**Fig. 2 | Balance in the direction of valuable power flow. a** Consistency of high-low price direction (real-time market). Each point represents one link, and there are the following number of links in each category: cross-interconnect: 9, interregional: 31, within-region: 30. The horizontal lines on each box plot show, from low to high, the smallest data point lying within 1.5x the inter-quartile range (IQR) from the 25th percentile, the 25th percentile, the 50th percentile (median), the 75th percentile, and the largest data point lying within 1.5x the IQR from the 75th percentile.

Supplementary Fig. 1 provides analogous information for the day-ahead market. **b** Example of a link-year with balanced value of power flow in each direction. The graph shows the distribution of marginal transmission market value (real-time market) across the months of 2022 and by which location had the higher price in any particular hour, for the link between southern Texas and Louisiana. For this example, the maximum share of value in a single direction – the metric used in (**a**) – is 0.54.

upgrades have nearly eliminated congestion costs in the New England energy market"[40].

Persistent pricing gradients often underlie this value, but do not fully capture it since the lower-priced location of the pair can alternate over time. The transmission value suggested by differences in annual average prices is typically only 18–45% of the value suggested by our methodology based on hourly differences in real-time prices for cross-interconnect links, 26–61% for interregional links and 35–81% for within-region links. Similarly, only considering the value in one direction (e.g., when the price on side A of the link is higher than on side B) captures just 59–67%, 63–80% and 67–90% of the value for cross-interconnect, interregional and within-region links, respectively. All ranges reported in this paragraph reflect the 25th-75th percentiles of

the studied transmission links. As shown in Fig. 2a, cross-interconnect and interregional links tend to be fairly balanced in terms of which direction power flow would be valuable over time. Figure 2b shows an example of a balanced link in a specific year. Studying directionality has important implications for concerns over winners and losers in transmission planning[41,42]. Balanced directional value could suggest comparable benefits from transmission development for multiple parties.

### Temporal trends in transmission value

The market value of transmission varies over the course of years, but also between the day-ahead and real-time markets and from hour to hour. The bars in Fig. 3 show the trend in transmission value for each year 2012–2022. In 2022, large locational price differences were a

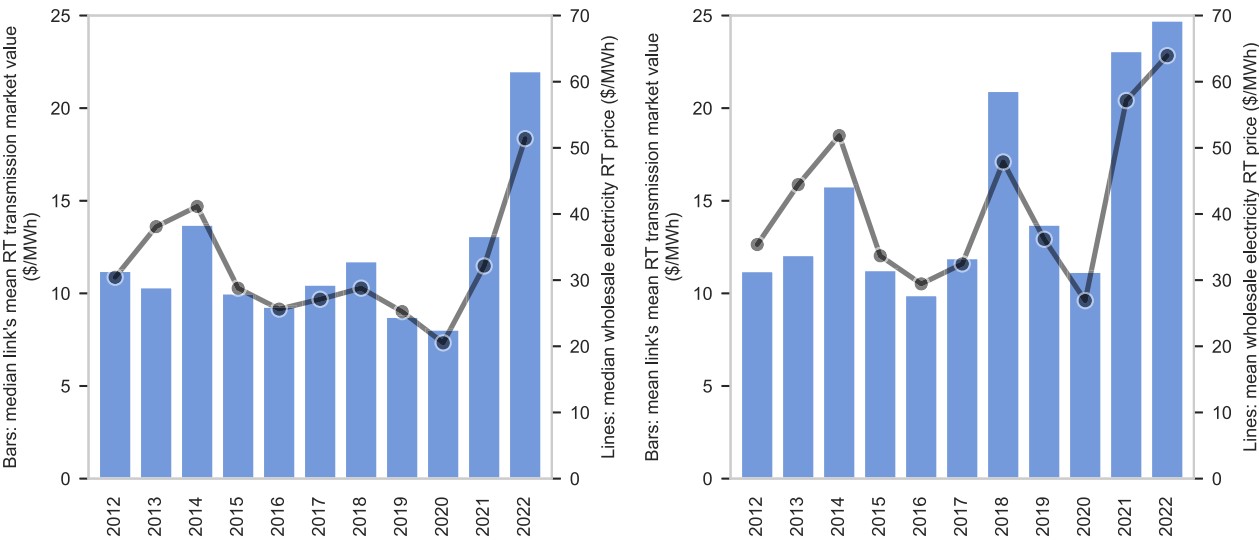

**Fig. 3 | Median (left) and mean (right) real-time market value of transmission (bars) and wholesale electricity price (lines) across the set of 70 links.** Note that the set of links in the early years is smaller due to data constraints, as explained in the Methods. Supplementary Fig. 2 provides analogous information for the day-ahead market.

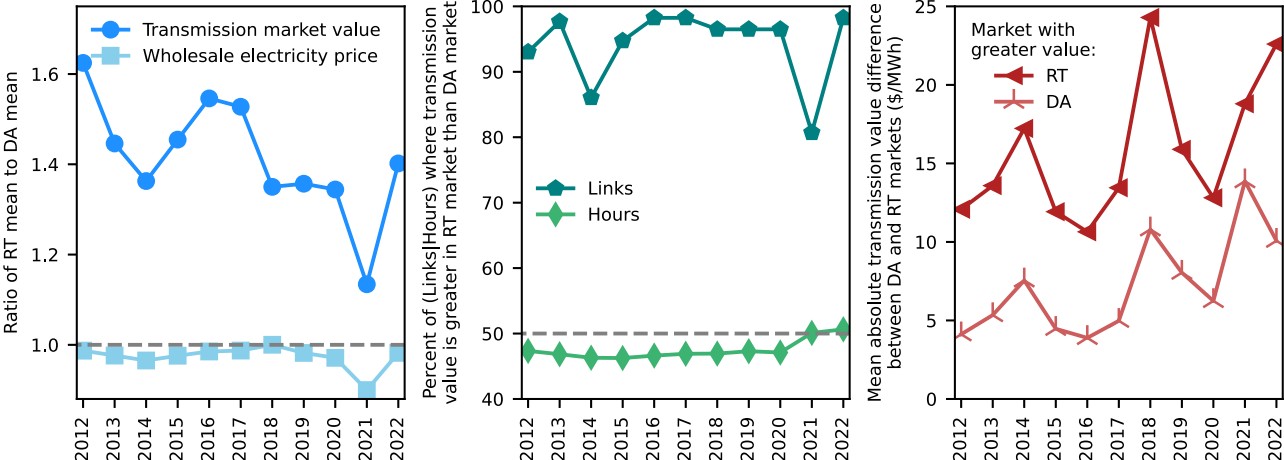

**Fig. 4 | Comparison of transmission market value in day-ahead and real-time markets.** Excludes links connected to the non-ISO West where there is not a day-ahead market. The set of links in the early years is smaller due to data constraints. Left: Average in the real-time market relative to average in the day-ahead market for (1) transmission market value and (2) wholesale electricity prices. Center: Prevalence of greater real-time transmission value in terms of (1) the share of links where annual real-time value was greater than day-ahead and (2) the share of hours across all links in which real-time value was greater than day-ahead. Right: Magnitude of the average hourly difference between day-ahead and real-time transmission value presented separately for hours in which (1) real-time value was greater than day-ahead value and (2) day-ahead value was greater than real-time value.

broad phenomenon across most of the U.S., resulting in the highest median (left) and mean (right) transmission value of any calendar year since at least 2012, the earliest year in-scope of this study. The increase in transmission value across so many locations is suggestive of a cause that is national in scope, such as overall increased energy prices (see the lines in Fig. 3). In contrast, high mean values without corresponding high median values, such as in 2018 and 2021, indicate events that drove extremely high transmission value in isolated regions. In 2021, for example, winter storm Uri drove high values for interregional transmission into SPP and ERCOT but had less impact on other regions of the U.S. This pattern underscores the importance of considering a long time horizon when analyzing transmission investments, which have lifespans of 50-80 years, while also considering possible value drivers that derive from national, regional, and local conditions.

**Comparison of market timeframes.** Focusing on shorter, operational time scales, a typical link had a transmission value 30% greater in the

real-time market than in the day-ahead market. While electricity prices are positively correlated with transmission market value across years, the same is not true when looking at day-ahead and real-time markets: Real-time transmission value is greater than day-ahead value on average (Fig. 4, left panel) and for the vast majority of links each year (Fig. 4, center panel), yet average electricity prices are lower in the real-time market (Fig. 4, left panel). It is the magnitude, not frequency, of the change in transmission value between markets that primarily drives greater real-time value: Our analysis finds that a link's market value is approximately equally likely to increase or decrease from the day-ahead to the real-time market for any given hour (Fig. 4, center panel), but the mean increase is far greater than the mean decrease (Fig. 4, right panel). Because an increase in transmission value often coincides with a price increase at one or both nodes, this finding is consistent with the real-time nodal price(s) being on more inelastic portions of the supply curve(s) when transmission value increased relative to day-ahead than when it decreased. Transmission value is also more volatile

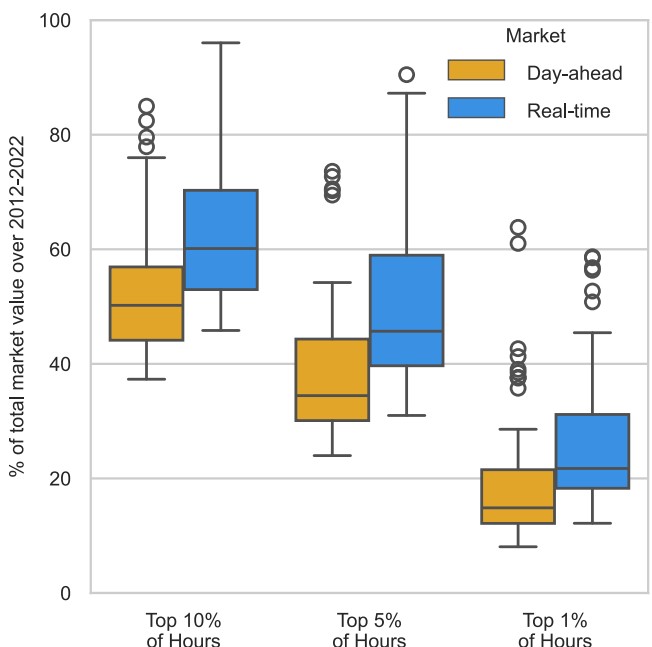

**Fig. 5 | Transmission's marginal market value was concentrated in a small fraction of the time period.** Sample sizes: RT market: 70, DA market: 57. The horizontal lines on each box plot show, from low to high, the smallest data point lying within 1.5x the inter-quartile range (IQR) from the 25th percentile, the 25th percentile, the 50th percentile (median), the 75th percentile, and the largest data point lying within 1.5x the IQR from the 75th percentile.

in the real-time market, with a standard deviation of $42/MWh compared with $16/MWh in the day-ahead market for a typical link.

**Transmission value is concentrated in time.** Given the volatility of congestion, aggregating to calendar year intervals offers an incomplete view of how a link's value potential is distributed over time. Instead, we consider how the value is distributed within the entire study horizon. This perspective is presented in Fig. 5 and reveals that, for real-time markets, at least 45% of total value is accounted for by only 10% of hours for all links and by just 5% of hours for the majority of links. Narrowing further, the top 1% of hours by real-time market value typically account for 20–30% of a link's potential, but it can reach over 50% in some cases. Transmission value in day-ahead markets is not only lower than in real-time markets, as discussed earlier, but also more dispersed. Figure 5 shows that a smaller share of total day-ahead market value is concentrated in the hours with the highest day-ahead value as compared to real-time. Still, transmission value in day-ahead markets is not distributed evenly across all hours and the majority of day-ahead transmission value is usually found in just 10% of hours. Since transmission's marginal market value is heavily influenced by a relatively small collection of key hours with exceptionally large price differences, capturing the tails of market outcomes is important for transmission planners aiming to estimate the value of a transmission line.

**Conditions during times of peak transmission value**
To explore conditions during times of high transmission value and therefore assess potential drivers of these large geographic price spreads, in this section, we focus on the hours where the real-time transmission value is in the 95th percentile or above for each link and refer to these 5% of hours as the peak value hours. Due to net load data availability and the absence of a day-ahead market in the non-ISO West, this section focuses on the 52 studied links within or between ISO or RTO regions, excluding those in the non-ISO West and Southeast.

Unforeseen intraday variance, high net load, cold weather, or high renewable power generation conditions are present in over 75% of the hours with peak transmission value. This is not simply because these four conditions are pervasive and present most of the time. Rather, these conditions disproportionately coincide with the peak value hours.

Unforeseen intraday variance is identified by a large change in the LMPs between the day-ahead and real-time markets on either side of the link (see Methods for a precise definition). Such a change between markets reflects sizeable forecast errors in the day-ahead timeframe, for example, errors in estimated supply, demand, weather, or infrastructure outages, and limited flexibility to adapt to unexpected operational circumstances. These large price changes are a price increase 64% of the time and a price decrease the other 36% of the time. Of the transmission value resulting from peak value hours, 74% overlaps with a detected unforeseen event, as shown in Fig. 6a (i.e., the entire blue bar covers 74% of the left gray bar). Remarkably, 43% of all hours with an unforeseen intraday variance detected are in the 5% of hours with greatest transmission value. Figure 6b(i) shows how the distribution of hours with an unforeseen intraday variance are skewed toward the times of highest transmission value. This result directionally supports the simulation-based conclusion in ref. 18 that the benefits of adding transmission depend on the uncertainty between day-ahead scheduling and real-time operations. Unforeseen intraday variances sometimes occur during times of high net load, high renewable power generation and/or cold weather: Of the peak transmission value coinciding with large day-ahead to real-time price differences, 43% overlaps with one or more of these conditions, as shown by the lighter blue labeled segments in Fig. 6a.

Cold weather−defined here as the coldest 5% of days in each location – overlaps with 25% of the value resulting from the peak hours (Fig. 6a). Often this value overlaps with unexpected events, net load, or both, but cold weather is the only studied driver present for 5% of the peak transmission value (dark orange segment). High net load – defined here as the 5% of hours in each location with the highest net load – is rarely the only studied condition affecting a peak value hour (dark yellow segment; 2% of peak value). However, it is still an important factor because 20% of the value resulting from the peak value hours comes from times of high net load (Fig. 6a). High net load is the result of high electricity demand and/or low variable renewable energy generation. The distributions of high net load and cold weather periods in Fig. 6b(ii,iii) show a clear skew toward the highest value hours, though to a lesser degree than was observed for unforeseen intraday variance. While a disproportionate amount (13%) of cold weather and high net load hours coincide with the peak value hours, it is still much more common for a period with one of these conditions to pass without encountering a peak transmission value. Thus, peak value periods are difficult to predict, even if you are expecting very cold weather or high net load.

While high levels of renewable generation make a period less likely to have high net load, it can still be a driver of peak transmission value due to geographically varying renewable penetration rates and weather patterns. Focusing on the 5% of hours with greatest combined wind and solar generation in each region, Fig. 6b(iv) shows that hours with high renewable generation coincide more often with high transmission value than low value, but they do not have the same presence in the very highest transmission value hours as the other conditions discussed so far. Still, high renewable generation impacts a portion of peak transmission value which unforeseen intraday variance, high net load, and cold weather do not (green segment in Fig. 6a; 2% of peak value).

Supplementary Table 1 details the prevalence of each condition discussed here and its relationship with the top 1%, 5%, and 10% value hours. Further, the Supplementary Information includes analysis on the roles of hot weather and designated storms or grid reliability

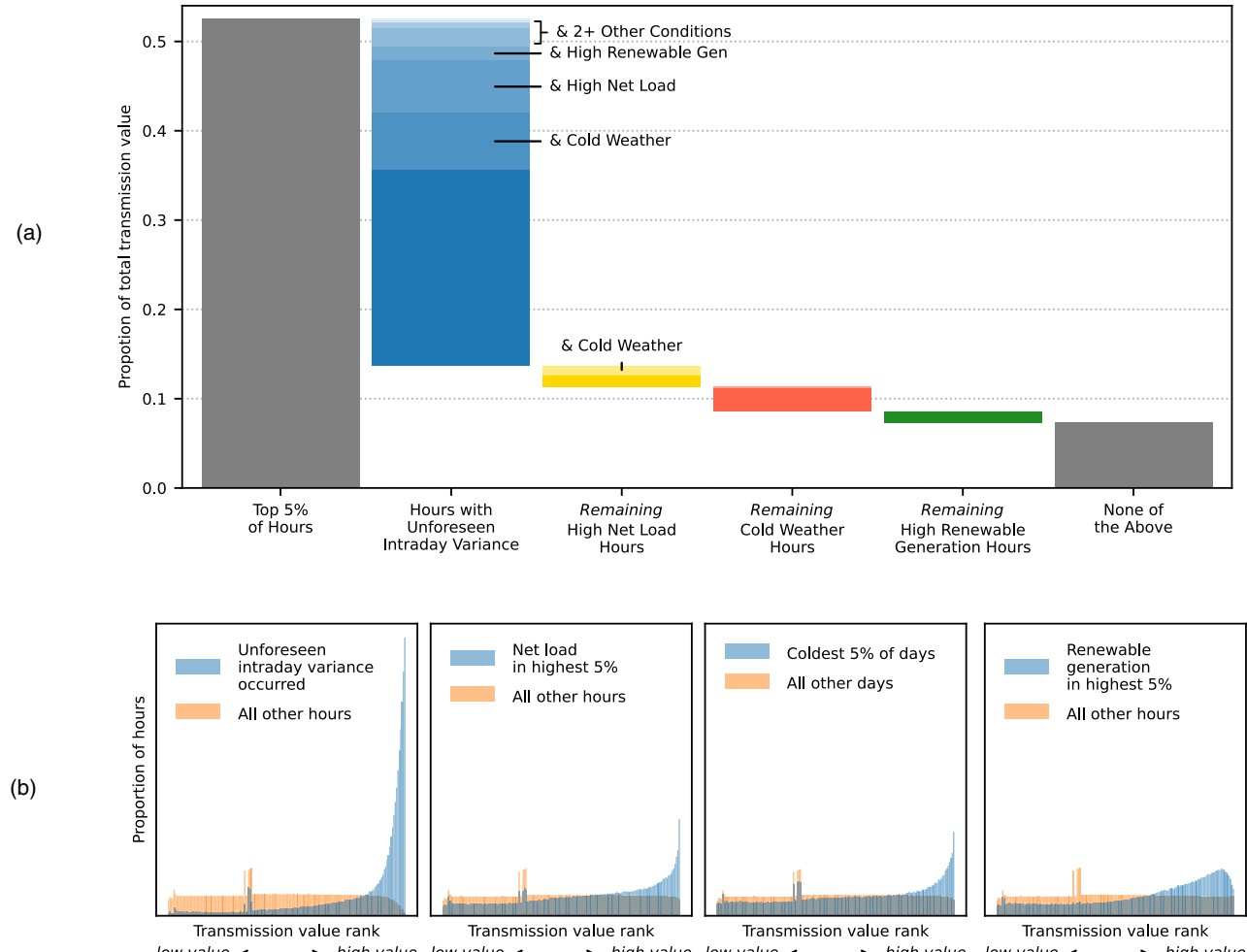

**Fig. 6 | Relationships between four key conditions and peak transmission value (real-time market).** Aggregate results for all 52 studied links within or between ISO or RTO regions, excluding those in the non-ISO West and Southeast. **a** Contribution of key system conditions to transmission market value during peak value hours (top 5%). This figure should be read from left to right. When multiple conditions are present at the same time, the associated value appears in the first (i.e., leftmost) applicable column and is identified by lighter segments and "& [overlapping condition]" labels. **b** Distributions of hours relative to transmission value when a key condition is present (blue) compared to when it is not (orange). The local peaks in the left half of each distribution represent a mass of hours each with $0 transmission value and therefore the same value rank for each link.

events on peak transmission value, which were found to be nearly subsumed under the four conditions analyzed here.

## Geographic differences in conditions during times of peak transmission value

The conditions analyzed above do not affect all links equally. Some conditions are more relevant for interregional links or within region links and others have an outsized impact on certain regions of the country. Figure 7 shows the distribution of links in terms of the share of peak value with each condition. In general, interregional and cross-interconnect links see a greater share of peak transmission value explained by these conditions. This is consistent with greater weather differences across longer distances and our use of regional-level load and renewable generation totals; local condition differences that are not investigated here may have more impact on within-region links. Unforeseen intraday variation is a key condition for all studied links, but especially for interregional links. Interregional transfer quantities are often scheduled several hours or more in advance, with less flexibility to adjust compared to within-region flows. These results point to the lack of interregional operational flexibility within 24 h of operation as a driver of transmission value. The two links with the greatest share of peak value during cold weather are SPP South < > ERCOT West and

MISO < > ERCOT, but all ISO/RTOs other than CAISO are connected to at least one link where cold weather is present for at least 35% of peak value. High renewable generation rarely coexists with peak transmission value for some links, but for some regional and interregional links, particularly those within or between MISO and SPP, there is considerable overlap.

Case studies investigating differences in the conditions affecting transmission value for two links in more detail are provided in the Supplementary Information, including in Supplementary Figs. 3 and 4.

## Market depth and saturation effects

This analysis has so far focused on transmission's marginal market value, which is defined precisely by market-clearing prices. If one is instead interested in the energy market value of a finite, larger transmission capacity addition, under the counterfactual that the line existed during 2012–2022, there are several factors to consider in addition to the observed marginal market value. One such factor is market depth; each increment of transmission capacity has the ability to reduce the price difference across its span, leading to an average value for the line that is less than the marginal value without it. The impact of market depth is accounted for in the following analysis through the use of statistically-based rolling supply curves and

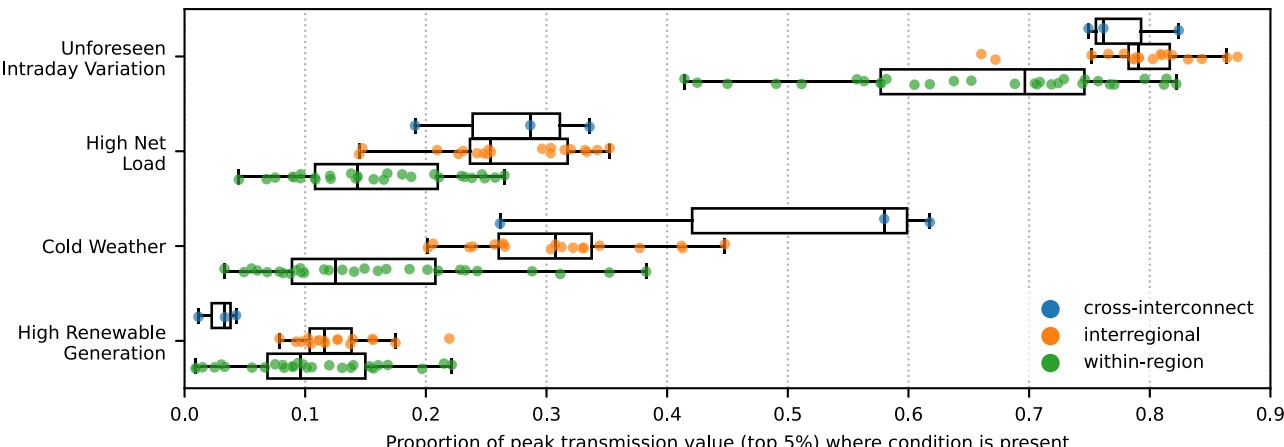

**Fig. 7 | Distribution of key conditions' impacts on peak transmission value (real-time).** Each point represents one of the 52 studied links (3 cross-interconnect, 19 interregional, 30 within-region) within or between ISO or RTO regions, excluding those in the non-ISO West and Southeast. The three cross-interconnect links are between ERCOT and either SPP or MISO. The horizontal lines on each box plot show, from low to high, the smallest data point lying within 1.5x the inter-quartile range (IQR) from the 25th percentile, the 25th percentile, the 50th percentile (median), the 75th percentile, and the largest data point lying within 1.5x the IQR from the 75th percentile.

assumes a static market. That is, it does not account for generator entry and exit[29,30], demand changes, shifts in market power[31,32,43], or other dynamic effects that transmission expansion may create.

Here, we summarize the analytical approach that is detailed in the Methods section. First, we infer supply curves from the data by fitting a non-decreasing third-degree polynomial to the relationship between regional net load and market price at each node for every bi-monthly period during the study horizon, similar to ref. 44. Second, we screen the fitted supply curves to identify which models represent the data sufficiently well and which models produce large-magnitude residuals for a substantial share of the hours. Models that fail to pass the screening are not used to estimate transmission value. Third, the supply curve models and hourly data are analyzed for each link to establish hourly transmission value estimates. In most hours, both (net load, price) observations are near their respective supply curves and the difference between these two supply curves forms a transmission demand curve. The area under this transmission demand curve, yet above the horizontal axis, between zero and the additional transmission capacity is the transmission value estimate (see Fig. 8). When an observation is not near the supply curve or the corresponding supply curve did not pass screening, we use alternative approaches that lead to lower and higher value estimates. Finally, hourly values are summed over the study horizon.

Applying this approach, we estimate the transmission value accounting for market depth of a 1 GW increase in transmission capacity individually for each of the 40 interregional and cross-interconnect links in this study. Unlike the directly observable marginal transmission value metric, this value is an estimate for a hypothetical situation. 1 GW of transmission capacity represents a relatively small addition for some links, such as those between PJM and MISO North where there is already 21.7 GW of transfer capacity, and a substantial increase in other regions, such as between PJM and New York and between New York and New England where it represents a 50% increase[20]. We find values typically 72% to 93% (25th percentile of lower estimate to 75th percentile of higher estimate) of the marginal market value, as shown in Fig. 9. Based on these findings, we conclude that, at a realistic line size of 1 GW, saturation effects are not dominant and instead represent a discount to marginal transmission value that is occasionally moderate but typically modest. Supplementary Fig. 5 presents results for a more aggressive test of market depth that assumes the market size (as defined by net demand) is 50% smaller than the market used here. In that case, transmission value accounting

for market depth of a 1 GW capacity increase is typically 59% to 87%[2] of the marginal market value.

## Transmission costs compared to market value

To better contextualize our market value estimates, we compiled data on the costs to construct individual transmission projects across different regions of the United States. In total, the 26 recent and proposed transmission lines we evaluated correspond to over 87 GW of potential transfer capability (details for each individual line are provided in Table 1) and can be loosely matched, on a regional basis, to a subset of the transmission link values we estimate. Such value-to-cost comparisons should not be used to assess the full cost and value of any individual, specific transmission investment. However, the value-to-cost ratios we present in Fig. 10 do provide some indication of the net economic value of transmission development, focusing here solely on energy market value and excluding other possible benefits of transmission investment.

Figure 10 shows the value estimate accounting for market depth and the cost of geographically similar transmission projects for the 26 projects (22 unique links). The figure breaks out our results for the three types of links we study: cross-interconnect, interregional, and within-region transmission lines. Figure 10b shows the range of cost and value estimates in units of $/MW-yr and shows variation both across the types of links as well as within a given link. Across all 26 projects studied, only 3 projects had cost ranges that exceeded the value estimate range. Figure 10a converts the absolute cost and value estimates into a value-to-cost ratio. We find that all cross-interconnect projects have value-to-cost ratios that exceed 4, suggesting that energy market value alone could motivate transmission development to link these areas. All these lines are in fact being privately proposed by merchant developers. The failure and challenge of developing some of these lines imply non-economic constraints. For instance, merchant developers often must seek customers to subscribe to their proposed lines, but those customers can be utilities that might own and operate their own transmission lines and thus not want to participate in such a competitive development process. For interregional lines within a common interconnection, value-to-cost ratios are smaller (median value of 1.6) with all of the ten lines having value-cost ratios > 1. Within-region lines have the lowest value-to-cost ratios with four of the eleven lines assessed having value-to-cost ratios <1 and the seven others > 1. Supplementary Figs. 6 and 7 present results for sensitivity cases under different depreciation rates and where the value estimate range is

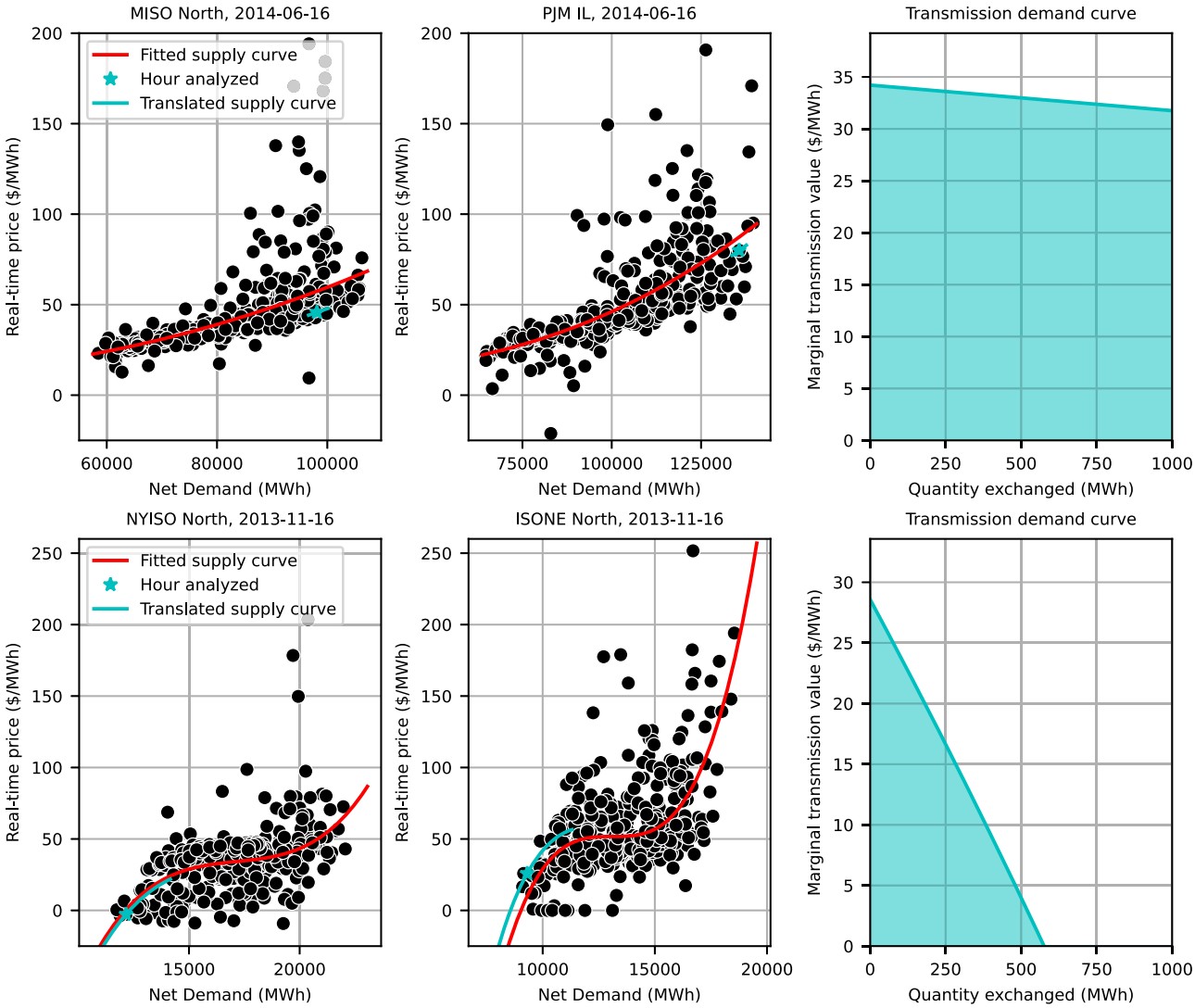

**Fig. 8 | Two illustrations of the methodology employed to account for saturation effects.** Left and center panels: Each black point is an hourly observation from the half-month period that starts on the listed date. The red curve is the supply curve fitted to these points, and the blue curve is the red curve shifted to pass through the hour currently being analyzed. Right panel: The blue curves in the left and center panels combine to the transmission demand curve shown here. The area under the curve is the transmission value for the specified hour. In the upper example, transmission value declines slowly and the value is close to the marginal value. In the bottom example, transmission value declines quickly and prices converge before 1 GW of capacity is fully utilized. Supplementary Fig. 9 offers three more detailed illustrations of the same methodology.

based on a market size (as defined by net demand) that is 50% smaller than the market used here.

Projects that have higher investment costs compared to historical market value may still have merit, especially if grid conditions are anticipated to change in ways that increase value over time. Additionally, energy arbitrage value is only one of many potential benefits of transmission: other drivers like renewable integration, electric reliability/resiliency, and avoided capacity investments may motivate investment and are not fully captured by wholesale energy prices. Furthermore, several of the lines we analyze came into service during the time period of market value estimation. Figure 10a shows that these lines tend to have lower value-to-cost ratios, especially for within-region projects. Such results imply that while market value still exists for those links, some of the market value may already have been absorbed by the completed transmission line.

## Discussion

For transmission, the number of merchant projects that have been built in a decentralized manner to capture the value of spatial arbitrage in U.S. electricity markets is small. For instance, in PJM, the largest electricity market in the U.S., only 1% of its total transmission capacity is made up by merchant lines[45,46]. This could be because all lines that would be valuable to the system are built centrally through regulated ISO, RTO, or utility planning processes. Alternatively, it could be because of barriers to capturing the value provided to the system and/or to financing and constructing the project. These results suggest the latter: that there is valuable transmission infrastructure not being built in a centralized or decentralized fashion.

We find that transmission links often have substantial economic value in their ability to increase energy trade and, in turn, reduce congestion. Interregional links (those crossing market or grid seams) are especially valuable and see fairly balanced trade in each direction, emphasizing the broad value of such investments and the importance of interregional planning that jointly accounts for benefits to multiple regions. Further, our scoping-level comparison of transmission infrastructure costs to historical energy market values finds greater value than cost for majority of links, including all cross-interconnect links. At the same time, the limited actual investment in interregional

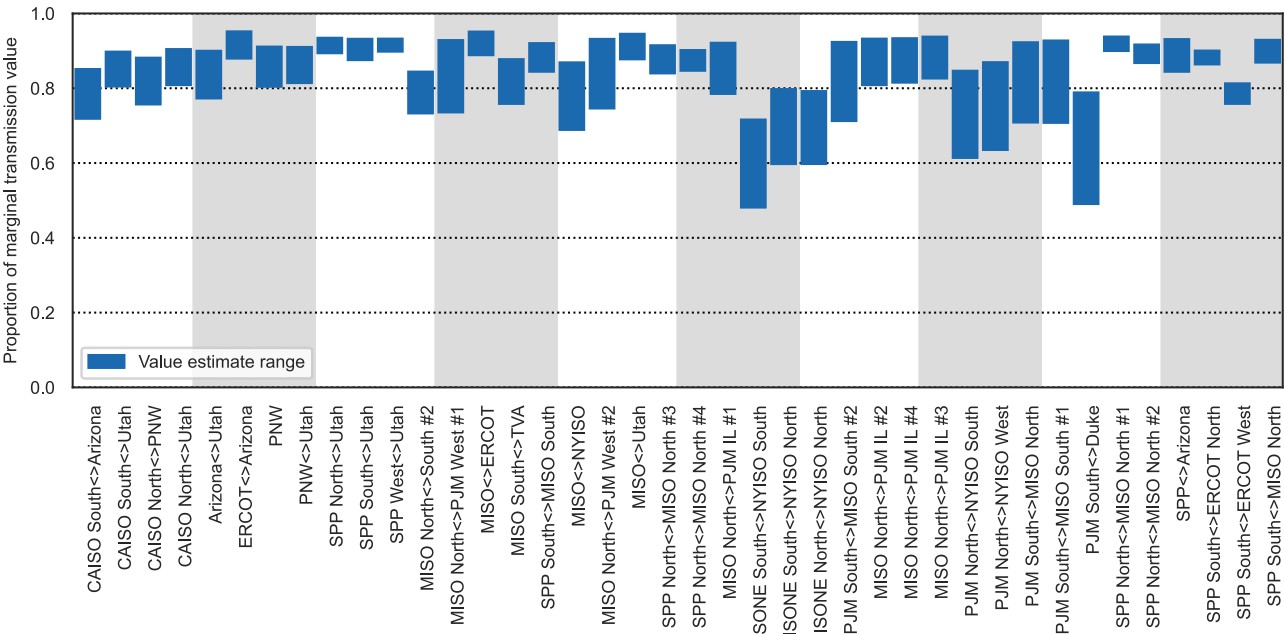

**Fig. 9 | Range of transmission market value estimates accounting for market depth of a 1 GW transfer capacity increase relative to marginal transmission value (real-time).** Gray and white bands are used to improve readability; they do not communicate information about the results.

transmission over the last decades suggests that barriers are effectively preventing development of otherwise valuable infrastructure.

Many types of barriers prevent an efficient buildout of transmission. For example, Joskow and Tirole[47], discuss how deviations away from perfectly competitive markets can interfere with efficient buildout of merchant transmission. Pfeifenberger et al.[19] add discussion on barriers presented by limited, localized transmission planning and the challenge of expanding both the size of the planning region and the types of benefits assessed during planning processes. FERC[8] focuses on barriers related to (1) regulatory review, including navigation of processes across different states, (2) limits to the locations at which transmission can be built, and (3) the length of time for new transmission to progress from plan to operation, which can take over a decade. The high value of transmission found in our work serves to highlight the cost of these barriers, that is, the potential net value is equal to missed savings. Each of the above citations discuss possible solutions to and implications of these barriers, including institutional reforms to planning, permitting, and cost allocation procedures.

Our findings also have implications for modelers, investors, planners, and policymakers and may serve to motivate and guide change to the processes through which transmission value is estimated. We find that the market value of transmission is highly influenced by the small fraction of time during which transmission is extremely valuable. In this sense, transmission can be thought of as having insurance value. Peak transmission value periods are primarily driven by events that are unforeseen or mis-forecast -12–36 h before the operating window. Secondary causes include high net load, cold weather, and/or high levels of wind and solar generation. Unforeseen intraday variance can occur due to demand and generation forecast errors, or unexpected outages of generators or transmission infrastructure. For modelers, improving the representation of these four drivers in planning scenarios may help improve performance. There is a wide variety of strategies for assessing these conditions in models. For example, models could benchmark simulated transmission market values against empirical market outcomes both in terms of their average magnitude and distribution of values over time. Additional research is warranted to help refine the types of conditions that have driven high transmission market value in the past, compare the results

of the empirical analysis in this paper with power-sector planning models, and to identify the highest priority areas for improvement.

For investors, historical pricing patterns can help identify areas where additional transmission investment may be warranted. Historical data should be paired with modeling simulations to properly judge cost-value tradeoffs over the expected 30+ year life of new infrastructure. However, many of the conditions that drove past transmission value are likely to persist for years to come, e.g., the benefits of weather, load, and generation diversity across large geographic regions. Historical patterns of transmission value can therefore be a useful complement to model results. For example, both forward looking and historical analysis were used by the United States Department of Energy (DOE) to develop a preliminary list of National Interest Electric Transmission Corridors[20,48]. These corridors are important because they allow transmission developers access to federal financing and permitting tools. The corridors reflect the key high-level findings in this paper, in that 8 of the 10 corridors are interregional, with 3 of those designed to facilitate trade across interconnections, and further, many of the corridors address high value links identified here.

For planners and policymakers, several important implications derive from our results beyond the aforementioned model improvements that will have derivative value to planners. First, and most simply, they suggest that barriers have been thwarting otherwise valuable transmission investment and thereby provide a signal to policymakers and planners that more effort is needed to identify and mitigate those barriers. Second, the results highlight where additional planning and policy effort may bear the most fruit. In the United States, recently finalized federal regulations have focused on generator interconnection to local transmission networks and within-region transmission planning (refs. 49,50). As has been noted by others (e.g., ref. 51), and as supported by our analysis, a critical next step is to address barriers to interregional and cross-interconnection transmission. Finally, among these barriers has been the perception that such investments primarily benefit consumers in one region at the expense of consumers in the other. On the contrary, the results presented here indicate that in many cases the benefits are bi-directional, and even more balanced for cross-interconnection and interregional lines. Such a finding should further motivate planners and policymakers to overcome the hurdles to new transmission investment.

**Table 1 | Summary of online and proposed transmission lines, the corresponding link name we associate, and the total cost and source**

| Project Name | Link Name | Line Capacity (MW) | Technology | Geographic Scope | Status | Business Model | Total Cost ($2022 Million) | Annualized Cost ($2022 million/yr) | Source |
|---|---|---|---|---|---|---|---|---|---|
| Tehachapi Renewable Transmission Project | CAISO_LosAngeles <--> CAISO_DesertHub | 4500 | AC | Regional | Complete | Utility | $3765 | $180 | Gorman (2019) |
| Sunrise Powerlink | CAISO_LosAngeles <--> CAISO_DesertHub | 1000 | AC | Regional | Complete | Utility | $2371 | $114 | Gorman (2019) |
| Ten West | CAISO_DesertHub <--> EIM_AZ | 2700 | AC | Interregional | Non-Complete | Utility | $302 | $14 | Grid Strategies (2023) |
| Grand Prairie Gateway | MISO_WI <--> PJM_IL | 1000 | AC | Interregional | Complete | Utility | $326 | $16 | Gorman (2019) |
| Eldorado-Ivanpah | CAISO_DesertHub <--> EIM_AZ | 1400 | AC | Interregional | Complete | Utility | $425 | $20 | Gorman (2019) |
| Plains & Eastern | MISO_LA <--> SPP_SouthHub | 4000 | HVDC | Interregional | Non-Complete | Merchant | $2880 | $138 | Gorman (2019) |
| Devers - Valley No. 2 Transmission Project DPV2 | CAISO_DesertHub <--> EIM_AZ | 1250 | AC | Interregional | Complete | Utility | $970 | $47 | Gorman (2019) |
| Transwest Express | CAISO_DesertHub <--> EIM_UT | 3000 | HVDC | Interregional | Non-Complete | Utility | $3456 | $166 | Grid Strategies (2023) |
| Pecos West Intertie | EIM_AZ <--> ERCOT_WestHub | 1500 | HVDC | Cross-Interconnect | Non-Complete | Merchant | $1500 | $72 | Online Webpage (2023) |
| North Plains connector | MISO_ND <--> EIM_UT | 3000 | HVDC | Cross-Interconnect | Non-Complete | Merchant | $3200 | $153 | Online Webpage (2023) |
| MISO Multi-Value Projects #3 | MISO_INHub <--> MISO_MIHub | 1486 | AC | Regional | Complete | Utility | $1358 | $65 | Gorman (2019) |
| MISO Multi-Value Projects #2 | MISO_MNHub <--> MISO_INHub | 3377 | AC | Regional | Complete | Utility | $2143 | $103 | Gorman (2019) |
| Southern Cross (Spirit) | MISO_LA <--> ERCOT_SouthHub | 3000 | HVDC | Cross-Interconnect | Non-Complete | Utility | $2700 | $129 | Grid Strategies (2023) |
| Clean Path New York | NYISO_NYC <--> NYISO_NorthHub | 3800 | HVDC | Regional | Non-Complete | Merchant +State | $3780 | $181 | Grid Strategies (2023) |
| Empire State Connector | NYISO_NYC <--> NYISO_NorthHub | 1250 | HVDC | Regional | Non-Complete | Merchant | $1765 | $85 | Grid Strategies (2023) |
| CREZ | ERCOT_WestHub <--> ERCOT_SouthHub | 18500 | AC | Regional | Complete | Utility | $8119 | $389 | Gorman (2019) |
| Southwest Minnesota Wind Expansion | MISO_SD <--> MISO_MNHub | 800 | AC | Regional | Complete | Utility | $395 | $19 | Gorman (2019) |
| MISO Multi-Value Projects #1 | MISO_WI <--> MISO_MNHub | 9137 | AC | Regional | Complete | Utility | $3596 | $172 | Gorman (2019) |
| Grain Belt Express | SPP_SouthHub <--> MISO_INHub | 5000 | HVDC | Interregional | Non-Complete | Merchant | $8065 | $387 | Gorman (2019) |
| Gateway West | EIM_BPAHub <--> EIM_UT | 3000 | AC | Interregional | Non-Complete | Utility | $4707 | $226 | Grid Strategies (2023) |
| Wyoming Intertie | EIM_UT <--> SPP_WestHub | 3000 | HVDC | Cross-Interconnect | Non-Complete | Merchant | $1000 | $48 | Online Webpage (2023) |
| SOO Green | PJM_IL <--> MISO_MNHub | 2100 | HVDC | Interregional | Non-Complete | Merchant | $4000 | $192 | Grid Strategies (2023) |
| Gateway South | EIM_AZ <--> EIM_UT | 3000 | AC | Interregional | Non-Complete | Utility | $4707 | $226 | Gorman (2019) |
| SPP Priority #1 | SPP_SouthHub <--> SPP_WestHub | 2625 | AC | Regional | Complete | Utility | $1059 | $51 | Gorman (2019) |
| SPP Priority #2 | SPP_KS <--> SPP_NorthHub | 375 | AC | Regional | Complete | Utility | $376 | $18 | Gorman (2019) |
| Three corners Connector | SPP_SouthHub <--> EIM_AZ | 3000 | HVDC | Cross-Interconnect | Non-Complete | Merchant | $2000 | $96 | Online Webpage (2023) |

## Methods

### Methodology foundation: market price signals

In electricity markets that employ spatially-differentiated marginal pricing (i.e., locational and, to some extent, zonal pricing), prices signal where investment is needed in the system. High prices incentivize the deployment of lower-cost generation resources and energy efficiency measures. Large variations in prices over time at the same node provide incentives for demand response and energy storage investments. Concurrent price differences between nodes indicate network congestion and signal that additional transmission capacity would be valuable. A price difference between two nodes could be caused by congestion exclusively on a line directly connecting them, but it

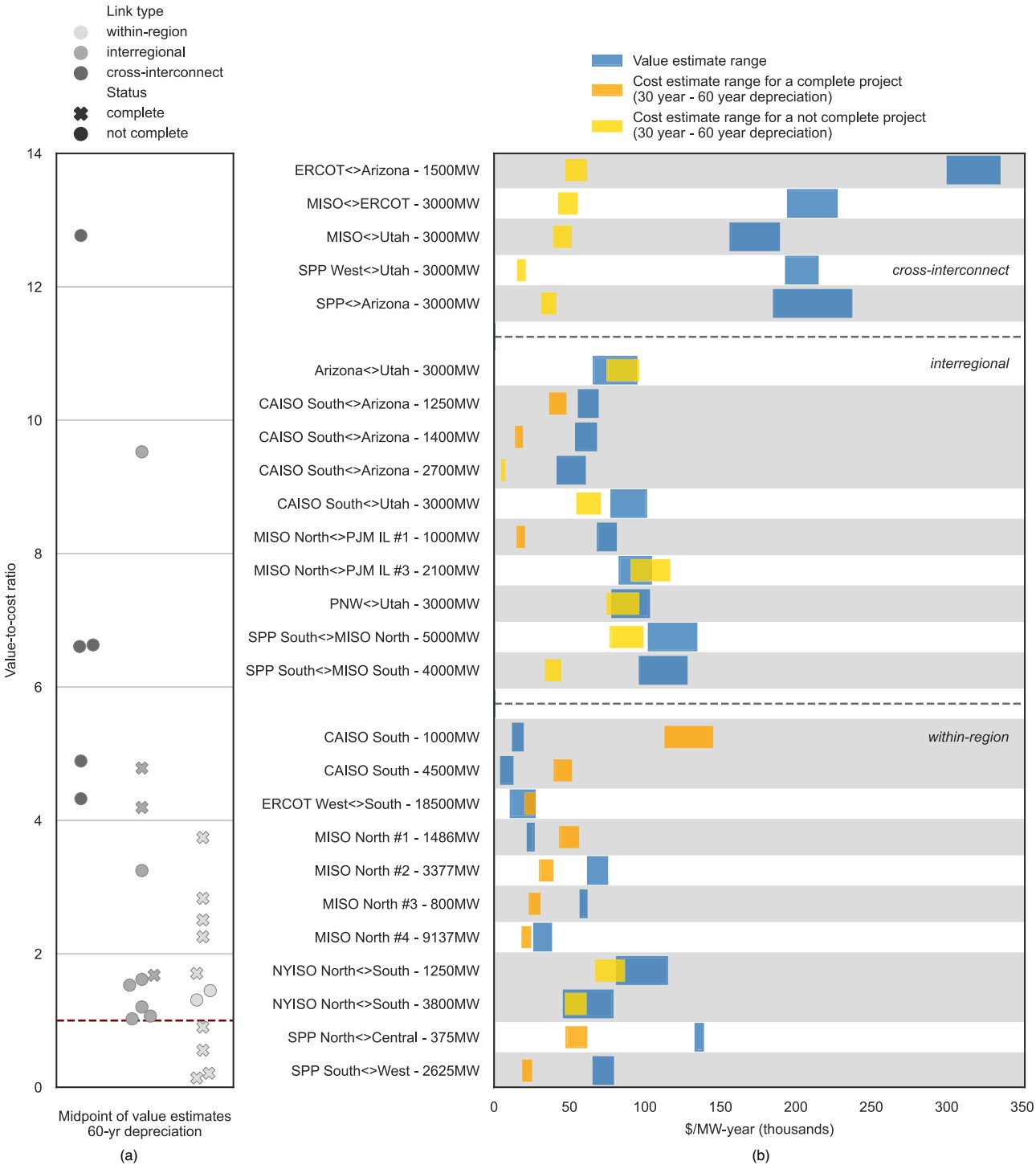

**Fig. 10 | Comparison of project costs to market value estimates that account for market depth.** See Table 1 for details on the project cost data. See Supplementary Figs. 6 and 7 for versions of this figure comparing multiple value and cost assumptions. **a** Value-to-cost ratios where the value is the midpoint of the value estimate range in (**b**) and the cost estimate assumes 60-year depreciation. The horizontal dashed line at 1 represents the break-even ratio. **b** Range of market value (real-time) estimates accounting for market depth and annualized project cost estimates.

typically is the result of simultaneous limits on multiple pieces of transmission infrastructure. These market signals of transmission value are the focus of this paper. While markets with a single market clearing price (e.g., Germany) provide location-agnostic signals for some of these investment types and indicators of transmission value may be visible to system operators (e.g., as shadow prices in optimal dispatch models), the presence of market-facing signals reflecting the

value of specific transmission capacity is unique to markets with spatially-differentiated marginal pricing (e.g., the liberalized electricity markets in the U.S.: CAISO, ERCOT, ISO-NE, MISO, NYISO, PJM, SPP)[52].

**Marginal transmission market value based on historical data**
This study utilizes wholesale locational marginal price differences as a market signal for the value of additional transmission between two

locations. The following equation defines the transmission market value quantity used throughout the paper,

*Mean marginal transmission market value between nodes A and B*

$$= \frac{1}{n} \sum_{t=1}^{n} |\text{price}_t^A - \text{price}_t^B| 1 \qquad (1)$$

*Equation 1: Definition of the mean marginal transmission market value between two nodes from time $t = 1$ to time $t = n$.*

where $n$ is the number of hours under consideration, the market price at node $X$ during hour $h$ is denoted $\text{price}_h^X$, and $|\cdot|$ is the absolute value operator. The marginal transmission market value at a fixed time (i.e., $|\text{price}_h^A - \text{price}_h^B|$) is sometimes referred to as the locational spot price of transmission[53].

These are real economic signals, but they are distinct from other measures of transmission's economic impact to the system and should be interpreted with care. One interpretation is that the mean marginal day-ahead transmission market value between nodes A and B is equivalent to the average hourly payment from holding two financial transmission right (FTR) options of equal size over the entire study period: one from A to B and one from B to A. FTR options are financial instruments that pay the LMP difference between two nodes if that difference is positive[28]. Another interpretation is that, if a link were an existing transmission line, the average congestion (i.e., shadow) price of the line's transmission constraint would equal the link's mean marginal transmission market value.

If a new transmission line were to have been built by 2012 between two nodes linked in this study (hypothetically), the mean marginal transmission market value multiplied by the number of hours is the marginal impact of the line on the total cost of energy bought by loads. The total change in energy costs for a static market is at most the line's marginal impact multiplied by its rated capacity for several reasons, including: (1) The transmission market value is marginal. That is, the absolute nodal price difference with the line is typically less than the price difference without the line and so, while the first unit of incremental transmission reduces costs at the rate of the transmission market value, the line as a whole impacts energy costs at a lower rate due to saturation effects. This is addressed in the next section. (2) The line may not be fully utilized even when there is congestion due to security constraints and power flow dynamics. The latter refers to the fact that the path of power flow on alternating current (AC) networks cannot be arbitrarily prescribed, because the electricity flows across transmission facilities according to Kirchoff's Laws based on the power consumption of loads and the voltage magnitude and real power injection established by generators. While direct current (DC) lines offer greater control over power flows, the surrounding AC system dynamics could still limit utilization. (3) Transmission expansion can further lower energy costs by increasing supply-side competition, reducing the ability of generators to exert market power[31,32,43]. On the other hand, while the marginal market value reflects elements of transmission's system value (e.g., savings to electricity production costs and the costs of ensuring reliability) it does not fully capture all benefits (e.g., impacts on generation capacity sharing, reduction in unpriced or mispriced emissions) and it includes generator rents, a reduction in which is not necessarily a net economic benefit but rather a redistribution. Further, there are dynamic impacts of transmission expansion on generator entry and[30] found that "ignoring this dynamic effect (…) would substantially understate the benefits of transmission investments." Therefore, a link's empirical marginal market value cannot be treated as a direct measure of, or strict upper or lower bound on, the total economic impact of expanded transmission capacity along the link. Still, we contend that market outcomes are a meaningful tool for measuring transmission value because concurrent price differences between market nodes indicate network congestion and serve as an investment signal to market actors and because

empirical prices precisely reflect actual system conditions and market participant behavior.

In our empirical analysis, the prices used are the observed market prices as established by the corresponding Independent System Operator (ISO), Regional Transmission Operator (RTO), or energy imbalance market. Real-time prices (averaged within each hour) are used as the default, since they most closely reflect the actual operating conditions of the system. Day-ahead prices are also used selectively as a point of comparison, and values based on day-ahead prices are clearly labeled as such. All results dependent on day-ahead prices exclude links connected to the non-ISO West, as the Western Energy Imbalance Market did not operate a day-ahead market over this period.

**Transmission market value accounting for market depth**

The locational spot price of transmission is the value of the first increment of additional transmission capacity. As more energy is transferred, the prices on either side of the link will begin to converge and may equalize, if there is sufficient transfer capacity. In this paper, we estimate the effect of such price convergence and the depth of markets on transmission value for each link individually. That is, we only consider the change in prices for two nodes at a time, and we assume each link is the only transfer capacity addition when analyzing it. We do not consider a case where the capacities of all, or a group of several, analyzed links are expanded simultaneously.

To account for the impact of market depth on transmission value, we first computed supply curve functions representing the relationship between regional net load and market price at each node over each bi-monthly (i.e., 1st–15th and 16th–end of month) period during the study horizon. Robust polynomial regression with shape constraints was used to find the best-fit cubic function that is non-decreasing on its domain, where best-fit is defined according to the Huber loss[54]. Each function's domain was defined as the range of net load values during the period extended in both directions by 10% of this range to allow for some extrapolation without encountering negative slopes. Huber loss uses the squared loss for samples with smaller residuals and the absolute loss for samples with larger residuals, making the model robust to outliers without ignoring their effect. The choice of period length and polynomial order was informed by ref. 44. Our Python implementation of this method was based on ref. 55. Demand is modeled as perfectly inelastic, because demand responsiveness is already reflected in the observations and therefore captured as a supply resource in the fitted supply curves.

Second, we screened the fitted supply curves to identify which models represent the data sufficiently well and which models produce large-magnitude residuals for a substantial share of the hours. Algorithm 2 in the Supplementary Information details this screening procedure. To summarize Supplementary Algorithm 2, we define an adaptive tolerance level for residuals in each bi-monthly period and a model passes the screening if at least two-thirds of the hours during the period are within the tolerance level. The tolerance level is always at least \$25/MWh and can be greater during periods of high price volatility, as determined by the median absolute deviation from the median price. Models that fail to pass the screening, such as the one depicted in Supplementary Fig. 8, are not used to estimate transmission value. For the 5256 total node-period supply curve models computed for the comparison of transmission value to cost, 99.8% passed the screening and at most 2 failed per node.

Third, the supply curve models and hourly data are analyzed for each link to establish hourly transmission value estimates. Here, we will describe the general methodology. In the Supplementary Information, Algorithm 1 details the complete procedure and Fig. 9 visualizes this methodology with several examples. For most link-hours – those with a model that passed the screening and a residual magnitude within the tolerance level described above on both ends of the link—a single value estimate is found. It is rare that the (net load, price) observation for an

hour falls directly on the supply curve, since the model represents the best fit over ~360 observations. The true market might behave similarly to the model at the observed net load or to the model at the observed price. So, the value estimate assumes prices at each node change at rates determined by the shape of each demand curve at the midpoint between these two points, i.e., the observed net load and the net load that the model predicts would produce the observed price. The difference between these two supply curve translations (e.g., the yellow curves in the left and center panels of Supplementary Fig. 9a,b) forms a transmission demand curve, and the area under this transmission demand curve yet above the horizonal axis between zero and the additional transmission capacity is the transmission value estimate. (This area is shaded in the right panels Supplementary Fig. 9a,b). If a node-hour has a successfully screened model but a large residual, we use two approaches to produce two estimates (see Supplementary Fig. 9c). The first approach simply uses a vertical translation of the supply curve; it assumes prices change as the model predicts around the observed net load. The second approach is more complex and assumes prices will change quickly with small changes in net load due to transmission. How quickly prices change is node-specific and determined using the steepest slopes found across all of the node's fitted models (excluding those that failed screening) on their observed net load domains, as detailed in Supplementary Algorithm 3. These are the steepest slopes across 264 models, for a node whose study period is 2012–2022. Finally, when a node-hour corresponds to a model that failed screening, we use only the latter, "steepest slope," method. This should produce a conservative value estimate in most hours because, typically, most hours in a 15-day period are on a relatively flat portion of the supply curve. Finally, any link-hours where prices are at or above the price cap for each node are treated as supply-constrained and assumed to have no trade on additional transmission capacity. Hourly value estimates for each link are aggregated into a lower estimate that is the sum of all singular hourly estimates plus the lower estimate when there are two estimates and a higher estimate that is the sum of all singular hourly estimates plus the higher estimate when there are two estimates.

**Conditions affecting transmission market value**
We analyze several environmental and market conditions that affect transmission value. Each of these conditions is defined based on a combination of measurements and criteria, as defined in Table 2. The unforeseen intraday variation criteria were designed to reflect large day-ahead-real-time price spreads while capturing a similar amount of time as the high net load, cold weather, and high renewable generation drivers. If a condition is present at one or both of a link's terminal nodes at a given time, we consider the condition to be present for the link.

**Transmission cost estimates**
A survey of large transmission project costs was taken from refs. 56, [57]. See Table 1 for details of the projects and corresponding cost estimates. These sources provide both a total cost estimate for construction of the lines (converted to 2022 dollars with the Bureau of Economic Analysis price deflators) as well as the total transfer capacity of the lines (in MW). However, they ignore operational costs for managing transmission lines. We apply the below equation to convert these capital cost estimates to annualized values to facilitate comparison to our transmission market value estimates.

$$Annualized\ cost\ of\ transmission\ investment = \left[\frac{(C*r)}{([1-(1+r)^{(-n)}])}\right] \div K$$

(2)

Where
 $C$ = capital cost of transmission investment ($2022)
 $r$ = real interest rate (%)
 $n$ = transmission asset lifetime (years)
 $K$ = incremental capacity of transmission infrastructure (MW)

Similar to ref. 56, we assume either a 60- or 30-year transmission line lifetime and a 4.44% real interest rate based on a 55/45 debt to equity ratio, 3.6% debt cost, 11.3% ROE, 26% tax rate, and 2% inflation rate. There is uncertainty in the transmission costs identified here, especially for projects that have only been proposed and not completed, as they may incur cost overruns, not represent the full all-in project cost, or not reflect differences in construction schedules.

Finally, we select a specific link from our market value analysis to correspond, as best as possible, with the geographic scope of the actual transmission line path. To perform this linkage, we compare the key market hubs used in our market value analysis to the planned electric linkages created by our actual transmission projects. This process is typically straightforward for cross-interconnect and inter-regional lines, where transmission projects will explicitly state their plan to electrically link two distinct market regions, oftentimes with HVDC transmission technology. This process is less straightforward for within-region lines, where the inherent nature of the AC connection makes it more difficult to assess transfer capability of new lines.

**Table 2 | Description of data and criteria used to test for the presence of key system conditions**

| Condition | Relevant measurements | Measurement granularity | Criteria |
|---|---|---|---|
| Unforeseen intraday variation | (a) Magnitude of the price change between day-ahead and real-time markets ($/MWh) (b) Magnitude of the price change between day-ahead and real-time markets as a percent of the day-ahead price magnitude (%) | Nodal | (a) $\geq \$40/MWh$ AND (b) $\geq 50\%$ |
| High net load | Electricity load less generation from wind and solar (MWh) | ISO/RTO | Top 5% of hours for that node within the study period |
| High renewable generation | Sum of generation from wind and solar (MWh) | | |
| Cold weather | Effective heating degrees (i.e., the difference between the measurement location's neutral heating temperature and the day's mean temperature) (°F) | ISO/RTO OR regional for ISO/RTOs with very wide geographic range | Top (coldest) 5% of days for that node within the study period AND Effective heating degrees > 0 |
| Hot weather | Effective cooling degrees (i.e., the difference between the day's mean temperature and the measurement location's neutral heating temperature) (°F) | | Top (hottest) {1%, 5%} of days for that node within the study period AND Effective cooling degrees > 0 |
| Designated events | NERC-identified periods of grid stress and key weather events identified in the literature (e.g., named storms, heatwaves). A complete list of events and their dates is available in Supplementary Table 2. | National | One or more designated events occurred that day |

Neutral heating temperatures are typically 60°F-70°F; see ref. 58 for more on heating and cooling degree days.

## Scope

Each pair of nodes analyzed is referred to as a link. 70 hypothetical transmission links are considered; 30 links are contained within a balancing authority ("within-region") and 40 links are interregional. Over 75% of pricing nodes at which links terminate are hub, zonal, or aggregate nodes that reflect the price signal for a group of buses in the same geographic area. Each of the remaining nodes is either a load, generator, tie generator, interface, or external node. The non-ISO Southeast nodes are of the latter two types as reported by MISO and PJM. Prices in the non-ISO West are Western Energy Imbalance Market prices reported by CAISO. Links between pairs of these nodes were established based on geographic relevance (e.g., a direct link between Arizona and Massachusetts was not included). Note that markets occasionally change their price node definitions over time. When the 2022 node does not exist for a given year, the geographically closest node of a similar type in the same balancing authority is used, if one exists. There are six nodes that experience a definition update during the study period.

The study period is 2012–2022 with the following exceptions: links involving SPP are 2015–2022 and the Pacific Northwest (PNW) link is also 2015–2022. All links have hourly real-time price data available for at least 90% of hours in each studied year and over 99% of studied link-years have at least 98% of data available.

## Reporting summary

Further information on research design is available in the Nature Portfolio Reporting Summary linked to this article.

## Data availability

We purchased data on market prices, load, renewable generation and weather from a commercial vendor; the product is called Velocity Suite, by Hitachi. This information is also publicly available.

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

## Acknowledgements

This material is based upon work supported by the U.S. Department of Energy's Office of Energy Efficiency and Renewable Energy (EERE) under Lawrence Berkeley National Laboratory Contract No. DE-AC02-05CH11231. The authors thank Patrick Gilman and Gage Reber of the Wind Energy Technologies Office and Paul Spitsen of the Strategic Analysis Team for supporting this work. The US Government retains, and

the publisher, by accepting the article for publication, acknowledges, that the US Government retains a non-exclusive, paid-up, irrevocable, world-wide license to publish or reproduce the published form of this manuscript, or allow others to do so, for US Government purposes.

## Author contributions

Julie Mulvaney Kemp: Methodology, Software, Formal Analysis, Writing – Original Draft, Review & Editing, Visualization; Dev Millstein: conceptualization, writing - review & editing, funding acquisition, supervision; Will Gorman: data curation, formal analysis, writing - original draft; Seongeun Jeong: data curation, visualization; Ryan Wiser: writing - review & editing, funding acquisition.

## Competing interests

The authors declare no competing interests.
