## [Transparent Peer Review file · Nature Communications]

Electric transmission value and its drivers in United States power markets

Corresponding Author: Dr Julie Kemp

Version 0:

Reviewer comments:

Reviewer #1

(Remarks to the Author)

Excellent paper with very interesting results. I certainly recommend the paper for publication.

Really appreciate several of the main points, including -- the idea that power flows in both directions on many ties; the market depth point to determine which lines are actually material; and the idea that much of the transmission value is in extreme hours.

I've also incorporated comments into the attached PDF, but a couple of items that you may want to address. (1) What percentage of the national benefit is attributable to ERCOT? Obviously a lot of low-hanging fruit there, but a separate political/planning discussion from the rest of the country. (2) I felt that a little more discussion about why the historical transmission cost data should be relied upon. Are the projects comparable size & scope? Same voltage? Etc. That type of analysis could be an independent paper, and I don't think it needs to be perfectly tied out, but a few more sentences on that issue would be helpful.

Anyway - nice paper and a pleasure to review.

(Remarks on code availability)

Reviewer #2

(Remarks to the Author)

The authors present an interesting analysis of transmission value and its drivers in the United States. This topic is very timely and of great interest to industry and researchers, given the growing interest in the role of transmission in current and future power systems, particularly in the context of the transition towards a low-carbon grid dominated by renewables. Below are some questions and comments to the manuscript.

- The analysis is based on a relatively simple, but insightful, investigation of historical prices. Could the authors elaborate on to what extent a similar approach has been used to analyze transmission value in the existing literature (including by the authors' own work)?
- In the introduction, it would be helpful if the authors could elaborate on the key contributions of the paper and how it relates to previous work. In particular, what is novel with this analysis?
- The authors state that the price-based analysis is not meant to provide model-based insights. What are the key aspects captured in this study, which is not addressed in model-based analysis. What are the main advantages of the approach? What value streams are not captured by market prices and how can they be assessed?
- Different metrics are used to describe transmission value (e.g. \$M/GW-year in abstract, \$/MWh in multiple graphs). Why?
- The authors use prices from the real-time market as the default. However, the quantity settled in the real-time market is typically much smaller than in day-ahead. Would not the day-ahead market provide a more robust benchmark?
- The term "unforeseen intraday variance" is rather generic and not directly linked to physical/weather parameters (as

opposed to the other event types as described in Table 1). How should we interpret this factor? What is cause and effect here? Where do criteria ($\geq \$40/\text{MWh}$, $\geq 50\%$) come from?

- The last paragraph on page 6 is hard to follow (e.g. "...are an increase 64%...") It is stated that 74% of peak value hours overlap with a detected event, but in the paragraph before it states that the events occur in more than 75% of peak hours.
- The font in figure 5 is too small and not really possible to read.
- The authors point out, correctly, the marginal nature of the analysis since it is based on historical marginal prices. The investigation of market depth and saturation is therefore an important addition. However, to what extent has the approach been validated? Did you attempt to measure the accuracy of the approach? Impacts on transmission expansion on prices is likely to be non-linear. Would non-linear impacts be captured? This should be elaborated in the paper. A brief description of the approach within the main text would be helpful for readability.
- In the first paragraph of the conclusion, what is meant with "frequent repricing instead of continuous repricing"? Also, the classical explanation of the missing money problem is price caps and possible other regulatory interventions that prevent the price to reach sufficiently high levels during supply scarcity.
- The authors state that the number of merchant lines built has been small in the US. Could you be more specific? How many? The authors allude to non-economic constraints as an explanation. What do you have in mind? Could it be that transmission value, the way it is calculated in the paper, is not a good proxy for investors (e.g. due to lumpiness of investments)?
- Have the authors compared their results to transmission value estimates from model-based studies? That would be a good extension that would link the analysis to other studies out there, and also add to validation of results.
- In methods, the authors claim that LMP markets are the only ones signaling the value of transmission. What about zonal markets?
- The authors should elaborate on the implications of the study. To what extent should these types of analysis be used to inform transmission planning (across the various types, including merchant)? Inform policy? Other uses?

(Remarks on code availability)

Reviewer #3

(Remarks to the Author)

This paper seeks to estimate the value (gross and net of construction and financing costs) of a large set of potentially "interesting" (looking backward) investments in a large set of specific transmission links using historical (2012-2022) hourly nodal price data, or more precisely differences in nodal prices, in RTO/ISOs in the United States to measure the gross value of each link. This analysis is combined with a set of data on the costs of building (investment and financing costs annualized investment costs over 30 or 60 years) the transmission links identified, if the links had been operating during this historical time period, to measure the net value of the link. The paper can be characterized as asking and answering the following question: "What would the value of this set of transmission links have been over the 2012-2022 period given the portfolio of generating plants in place during this period if the transmission links, one at a time, had been built prior to this period at the construction costs estimated in the paper?" The answer is that there are many links whose value, in terms of increasing generation from low-cost nodes over congested interfaces, would have more than covered the construction costs estimated in this paper. It is important to note that this paper focuses on a particular historical period and is quite different from the many analyses that have and are being performed in the U.S., the EU, Great Britain, and other countries that rely on optimization models to examine future scenarios, with an objective function with a larger set of values (e.g. includes carbon reduction benefits, portfolio evaluation to deal with uncertainty, etc.) and with fairly dramatic projected changes in the mix and location of generating plants and storage to meet GHG emissions reduction goals and the changing economics of generation and storage.

Comments and Questions

1. The paper is clearly written and the methods are presented in a reasonably clear and transparent fashion. The one section that I had trouble following is the section accounting for "market depth." Since this is a very important and complicated section the methods should be clear at least conceptually. More conceptual presentation should be in the text and not just in the methods section since this analysis is very important. I will discuss this section further below.
2. A very basic question is "Why does anyone care about answering the question posed above?" The past is the past. Investments take place in the future. We anticipate the generation mix and the distribution of nodal prices will change dramatically in the future as the types and location of generating facilities, storage facilities, and demand change in the future (EVs, heat pumps, data centers, etc.). The transmission facilities necessary to support this generation, storage and demand configuration are likely to be quite different from those made in the past or could have been made hypothetically to support historical generation and transmission configurations. This is why models are used to examine future scenarios and their attributes. As the authors note, the answer is that the value of the historical analysis is to demonstrate that barriers other than "economics" have kept socially valuable transmission links from being built in the past. I agree with this conclusion. However, the paper would have been much better if it had elaborated on what those barriers are likely to have been since these barriers likely continue to exist today and will create problems going forward as they have in the past. Relying on a very limited discussion of references is not enough.
3. I could not figure out whether the paper uses day-ahead or real time nodal prices to value transmission investments. Day-ahead prices are hourly forward financial commitments by generators and load to sell or buy specific quantities at the hourly market-clearing prices. (While these are forward financial commitments there are also physical implications, including unit commitment, max and min generation, congestion contingent bids, etc. A surprisingly large fraction of generators in the day-ahead market self-schedule rather than bid and are price-takers.) The real time market supports the physical dispatch of generators, storage, and price-sensitive demand to balance supply and demand and meet operating reliability criteria. The real time dispatch may vary from day-ahead schedules because demand is higher or lower than forecast, generating plants

have forced outages, more cheap imported energy may become available, etc. For example, upward deviations in generation from the day-ahead schedule and day-ahead price to meet dispatch instructions are compensated at the real time price for the increment in generation in excess of the day-ahead schedule only. Most generation and demand settles at the day-ahead price and from the perspective of the buyer and the seller at different nodes, the difference in the day-ahead prices between those nodes is the value of a (marginal) increase in transmission capacity between the two nodes. However, it appears that the paper uses real time prices to value transmission. It partially relies on a paper in a footnote whose analysis is not described. This section needs to be clarified and a simple couple of examples presented to justify using the real time prices if that's what they have done. In addition, it would be useful to present valuations using both day-ahead and real-time prices.

4. The authors have put together a very useful data set on transmission investments costs. However, two issues arise. First, the paper should demonstrate that the transmission cost data that are being used are representative of the "new" links being analyzed. The cost of a 20 mile upgrade of a 250kv line on an existing right of way may not be representative of a new 100 mile 500kv line on a new right of way. Second, a new link with a large transfer capacity is highly likely to require additional reinforcements at one or both ends of the AC system. For example, when the HVDC link from Quebec to Boston was built in the late 1980s, rather substantial investments in the New England AC network (NEPOOL at that time) were required to accept and disperse the additional energy injected at a new location outside Boston (an investment in PJM was also required for reliability). Failure to include these costs underestimates the true costs of the link.

5. I will turn next to the measurement of the effects of additional transmission capacity on nodal prices. This section is quite important for supporting the analysis. As a general matter, we expect that the difference in nodal prices will decline when a link is built that increases the capacity between the nodes. In the simple 2-node case, the nodal price at the low-price node with cheap generation should increase (unless there are constant marginal generation costs in which case the price would not change) and the nodal price at the high cost node with expensive generation should decrease as less costly but still higher-cost generation is required to meet demand at the high cost node (unless marginal generation costs are constant at this node too). However, if a lot of transmission capacity is added between the low cost and high cost nodes, the cheap generation at the low cost node could expand so much as to wipe out all generation at the high cost node. In this case, the difference in nodal prices would collapse to zero. In reality, this can be a very non-linear process --- the difference in nodal prices declines slowly as transmission capacity is added but if enough transmission capacity is added the price difference can suddenly collapse. A good transmission network model with existing generation located on it can perform this kind of "what if" analysis quite easily. It seems to me that this is more challenging in the context of this paper. The authors clearly do recognize that the nodal price only reflects the congestion cost associated with a small increment in transmission capacity. But the paper analyzes and finds profitable some very large lines. I understand the "market depth" analysis on a conceptual level, but "market depth" is not enough. We need supply and demand functions for each relevant market area and the entire ISO is unlikely to be a relevant market area since there is internal network congestion that yields smaller market areas. Anyway, the presentation in this section needs to be expanded and not rely only on the methods section. The issue can be presented with a couple of figures and then the key features of the analytical approach discussed in the text. The reader can then be sent to the methods section for even more details regarding each step.

6. I turn next to AC network issues. This paper is a transmission link by transmission link analysis. But AC systems obey Kirchhoff's laws. Accordingly, a large new transmission link can affect all of the nodal prices on a transmission network. This also has implications for AC network upgrades as I pointed out earlier. At the very least, the paper should at least mention this simplification and make the case that it is not important for the analysis.

7. Finally, I have one question and one suggestion. I could not figure out where day-ahead and real time nodal prices came from for the Southeast and Northwest since they do not have RTO/ISOs. Please make sure that the answer is clear. My suggestion is to consider comparing your results to the preliminary DOE analysis of National Interest Transmission Corridors ([https://www.energy.gov/gdo/national-interest-electric-transmission-corridor-designation-process.](https://www.energy.gov/gdo/national-interest-electric-transmission-corridor-designation-process)) and the DOE/NREL analyses of future transmission needs. Is the past prelude to the future?

(Remarks on code availability)

Version 1:

Reviewer comments:

Reviewer #2

(Remarks to the Author)

I appreciate that the authors have done a quite comprehensive update to the manuscript based on the reviewers' comments. I still have a few questions to the updated manuscript.

- Figure 4 is a nice addition, but could benefit from a more extensive description in the figure text. For instance, what does links and hours refer to in the middle panel?

- I still find the section on conditions during times of peak transmission value a bit hard to follow and suggest you make one more attempt to clarify the discussion around Figure 6. IN particular, why are the identified events (high net load, cold weather, high renewable generation) listed as separate factors as well as parts of the unforeseen events? I believe the reason is that they contribute to transmission value during classified unforeseen events and at other times, but this is not straightforward from the discussion. Why is cold weather written within the yellow bar for high net load? Does it occur at the same time as high load? If so, should high load be identified in the red bar for cold weather? In figure 6b, would it make

sense to also display the calculated transmission value?

- Figure 8 is a nice addition that makes it easier to understand the analysis around market depth. The sensitivity case with a 50% reduction in market size is also a good addition. The analysis still considers a 1GW expansion only. The new table 2 indicates that most transmission lines have a larger capacity. What would happen for a larger expansion of capacity, say 2GW or 5GW? Would Figure 9 look different in that case? Would the effect be similar to the case assuming reduced market size?

- Small detail: The end of the discussion/conclusion now uses the term cost assignment. I believe cost allocation is commonly used for this purpose.

- The new section on implications is a good addition to the manuscript

(Remarks on code availability)

Reviewer #3

(Remarks to the Author)

The responses to the referees' report are more than satisfactory. One item needs a bit more discussion. This is the table of projects that appears on page 15 of the response to referees and in the methods section. The body of the paper should make it clearer that more than half of these projects have not been completed so the costs are estimated costs. I have never seen a major transmission project that is completed at or below the estimated costs. So, I think that the actual costs are likely to be higher. (Some of these projects have faced permitting problems and may not even be built.) Also, the cost estimates are in different year dollars since the construction schedules are different. I think that this table should be in the body of the paper and not buried in the methods section, which almost nobody reads. This will make these attributes of the cost data clearer to readers. Finally, I am familiar with a few of these projects and, contrary to the statement in the response, they do not include estimates (or actual) of the full interconnection costs.

(Remarks on code availability)

Version 2:

Reviewer comments:

Reviewer #2

(Remarks to the Author)

Thank you for a careful revision. I do not have any further comments. Well done!

(Remarks on code availability)

Reviewer #4

(Remarks to the Author)

See Referee Report

(Remarks on code availability)

Version 3:

Reviewer comments:

Reviewer #4

(Remarks to the Author)

The authors addressed my comments.

(Remarks on code availability)

Review responses: Electric transmission value and its drivers in United States power markets

Summary note to all reviewers

We appreciate the engagement with our work from all three reviewers. We have found the comments constructive and have made an effort to respond to each of them thoughtfully. We believe the paper is improved over the first draft due to your feedback. The biggest updates we have made are 1) greater explanation of the approach to estimating value accounting for market depth, 2) a discussion section on “Implications for modelers, investors, planners, and policymakers,” 3) a table describing the transmission projects used for cost estimates, and 4) more substantial inclusion of day-ahead price analysis. Additionally, we have made numerous other edits to the paper and added additional content to the supplemental file.

Reviewer #1

What percentage of the national benefit is attributable to ERCOT? Obviously a lot of low-hanging fruit there, but a separate political/planning discussion from the rest of the country.

Indeed, links crossing from ERCOT into the eastern and western interconnects and within western ERCOT are some of the most valuable in our study. In general, we would caution against using these results to assign as specific percentage of the “national benefit” to a specific link or region. This is for a few reasons:

- 1) This is a link-by-link analysis, that is, we have not attempted to assess the value of expanding transmission capacity along all these links simultaneously.
- 2) The value of expanding transmission capacity on a link depends on how much capacity is added and the ideal capacity will vary across the map.
- 3) A complete national value would require greater visibility into the Southeast and West than we have provided (due to data limitations).

Still, using as a rough guide the total value assuming all capacity additions are of equal size and the marginal value is maintained over that entire capacity, the 8 links with one or both ends in ERCOT have 24% of the combined value in all 70 studied links.

I felt that a little more discussion about why the historical transmission cost data should be relied upon. Are the projects comparable size & scope? Same voltage? Etc. That type of analysis could be an independent paper, and I don't think it needs to be perfectly tied out, but a few more sentences on that issue would be helpful.

Thank you for this comment. Reviewer #3 had a similar concern. Please see our below response to Reviewer #3 and corresponding table where we provide more information about the historical transmission cost data that we relied upon in the paper. We've added this table to the methods section of the paper to provide more information on why the transmission cost data can be relied upon.

Reviewer 1 also provided detailed comments on the manuscript PDF, which we have responded to in kind and attached.

Reviewer #2

The analysis is based on a relatively simple, but insightful, investigation of historical prices. Could the authors elaborate on to what extent a similar approach has been used to analyze transmission value in the existing literature (including by the authors' own work)?

Sources [1] and [2] apply a similar marginal transmission market value methodology to estimating cost savings from additional transmission capacity during extreme weather events in the U.S. [1] includes a simplified method for accounting for saturation effects that only considers these effects in one of the two regions. [4] applies a similar marginal transmission market value methodology to Europe in 2019 and 2022. We, the authors of this paper, have used this marginal transmission market value methodology in [36] (a presentation slides-style report), [37] (policy brief), [38] (policy brief). None of these publications are in a peer-reviewed journal or conference proceedings. The introduction now includes the following text on the existing literature:

“Prior to this work, studies of transmission value using historical market price data focused on select severe weather [1], [2] or geopolitical [4] events, the specific benefits of increased competition [31], [32], measuring the impact of a specific transmission investment that is already online [28], [29], [30], or congestion on existing lines within a limited footprint (ISO/RTO annual reports, such as [33], [34], [35]). The authors of this paper have previously employed a subset of the methods used here and published the findings in [36], [37], [38].”

Relevant sources repeated here, for convenience:

[1] M. Goggin, “Transmission Makes The Power System Resilient to Extreme Weather,” Grid Strategies, LLC; American Council on Renewable Energy, Jul. 2021. [Online]. Available:

<https://gridprogress.files.wordpress.com/2021/11/transmission-makes-the-power-system-resilient-to-extreme-weather.pdf>

[2] M. Goggin and Z. Zimmerman, “The Value of Transmission During Winter Storm Elliott,” Grid Strategies, LLC; American Council on Renewable Energy, Feb. 2023.

[Online]. Available: <https://acore.org/wp-content/uploads/2023/02/The-Value-of-Transmission-During-Winter-Storm-Elliott-ACORE.pdf>

[4] George Dimopoulos, Conall Heussaff, and Georg Zachmann, “The massive value of European Union cross-border electricity transmission,” Bruegel. Accessed: Jan. 25, 2024. [Online]. Available: <https://www.bruegel.org/analysis/massive-value-european-union-cross-border-electricity-transmission>

[36] D. Millstein, R. H. Wiser, W. Gorman, S. Jeong, J. H. Kim, and A. Ancell, “Empirical Estimates of Transmission Value using Locational Marginal Prices,” Aug. 2022, Accessed: Oct. 10, 2024. [Online]. Available:

<https://escholarship.org/uc/item/5s42s6t7>

[37] D. Millstein, R. H. Wiser, S. Jeong, and J. M. Kemp, “The Latest Market Data Show that the Potential Savings of New Electric Transmission was Higher Last Year than at Any Point in the Last Decade,” Feb. 2023, Accessed: Oct. 10, 2024. [Online]. Available: <https://escholarship.org/uc/item/4sq3x22k>

[38] D. Millstein, J. M. Kemp, W. Gorman, and R. Wiser, “Transmission value in 2023: Market data shows the value of transmission remained high in certain locations despite overall low wholesale electricity prices,” Jul. 2024, Accessed: Oct. 10, 2024. [Online]. Available: <https://escholarship.org/uc/item/7rp5t6q6>

In the introduction, it would be helpful if the authors could elaborate on the key contributions of the paper and how it relates to previous work. In particular, what is novel with this analysis?

Thank you for identifying that the paper’s contributions were not clear in the introduction. The introduction now clearly states the purpose of this paper and how it compares to past studies of transmission value using historical market price data. As the introduction was extensively revised based on this comment and other comments, we will not attempt to reproduce here the specific changes, and instead refer the reviewer to the revised manuscript.

The authors state that the price-based analysis is not meant to provide model-based insights. What are the key aspects captures in this study, which is not addressed in model-based analysis. What are the main advantages of the approach? What value streams are not captures by market prices and how can they be assessed?

Thank you for this comment. We have added a subsection to the Discussion/Conclusion section, “Implications for investors, modelers, planners, and policymakers”, that addresses these questions – specifically it discusses what value streams may be missed in modeled based studies, and highlights key advantages of our approach. We believe this

work *does* provide general insights into the modeling process, especially identifying which types of processes in models may be causing biased results (we highlight treatment of unforeseen intraday variance specifically). However, as we state in the new subsection in the conclusion, further research is warranted to help identify the exact model processes that would have the largest impact on model results if refined.

A couple minor related points, it seems like this comment may be partly referring to sentence in the introduction ending with the clause "...we do not offer a one-to-one comparison between transmission's market value and its modeled benefit types." We have revised that sentence, as it was not clear. Finally, we note that in the introduction we state "While our approach is not meant to replace models, it is intended to help identify key mechanisms that may lead to biased model estimates of transmission value."

Different metrics are used to describe transmission value (e.g. \$M/GW-year in abstract, \$/MWh in multiple graphs). Why?

Good point, we have added content to the first paragraph of the section "Geospatial patterns of transmission value":

The marginal value is useful as it can be directly compared to market prices, but it can also be converted to an equivalent total value per link, such as millions \$/GW. Total value metrics are useful when compared to transmission costs. When considering total value metrics it is important to also consider market saturation effects (e.g., in each hour, is the full GW of transmission capacity needed, or would prices converge with less capacity?).

The authors use prices from the real-time market as the default. However, the quantity settled in the real-time market is typically much smaller than in day-ahead. Would not the day-ahead market provide a more robust benchmark?

We believe that day-ahead and real-time market signals of transmission value each can be relevant, depending on the questions you want to answer. To make the market differences more apparent to readers, we have expanded our paragraph on page 5 that compares day-ahead and real-time market prices and transmission value into its own subsection and introduced a figure visualizing the differences. Readers who have a preference for day-ahead transmission value can use the value ratios in the left panel of this figure as a rule-of-thumb for adjusting real-time values presented elsewhere. Additionally, we have added day-ahead versions of Figures 2 and 3 to the supplemental information. Ultimately, our decision to use real-time transmission value as our default comes down to three factors:

1. We are most interested in transmission's ultimate value to the system and real-time prices reflect physically binding dispatch decisions under the actual operating conditions of the system.
2. Past work (the source below, in particular) has identified real-time uncertainties as a critical driver of the benefit of geographic diversification that transmission provides. Given this literature, which is based on a modeling framework, we believe

it is important to use prices that reflect real-time uncertainties in our empirical assessment of transmission value.

- a) K. Van Horn, J. Pfeifengerger, and P. Ruiz, “The value of diversifying uncertain renewable generation through the transmission system,” Sep. 2020, [Online]. Available: <https://hdl.handle.net/2144/41451>
3. The Western Energy Imbalance Market (WEIM) is a real-time only market. So, if we focused on the day-ahead timeframe, we would need to either exclude these important links from the analysis altogether or use day-ahead prices for some links and real-time for the WEIM links, and we believe it is important to avoid “mixing apples and oranges” in this way as much as possible.

The term “unforeseen intraday variance” is rather generic and not directly linked to physical/weather parameters (as opposed to the other event types as described in Table 1). How should we interpret this factor? What is cause and effect here? Where do criteria ($\geq \$40/\text{MWh}$, $\geq 50\%$) come from?

You can interpret unforeseen intraday variance as something unexpected that occurs within 24 hours of the operating window at a time when the system has limited flexibility to adapt. This combination of factors (the cause) leads to big price swings between the day-ahead and real-time markets (the effect we observe). The “forecast error” in the day-ahead timeframe could be within attributes that are forecast in the traditional sense, like demand and weather, or it could be a forced transmission or generator outage. Because there are a large variety of these forecast errors that could lead to unforeseen intraday variance, we opted to identify them through their effect on prices, as opposed to collecting forecast and actual data on each forecasted attribute. That said, we do hope to explore some specific causes of unforeseen intraday variation in future work.

The specific criteria ($\geq \$40/\text{MWh}$, $\geq 50\%$) were designed to capture a similar amount of time as the high net load, cold weather, and high renewable generation drivers, which are each defined as the most extreme 5% of hours or days. We utilized both a relative metric and an absolute metric to avoid the following two types of DA-RT price spreads being included, as they don’t align with the intuition of a large intraday variance: \$1 to \$3 (i.e., large relative change, but small absolute change), \$450 to \$400 (i.e., large absolute change, but small relative change). We have added the following sentence to the “Conditions affecting transmission market value” section of the Methods to communicate this to readers:

The unforeseen intraday variation criteria were designed to reflect large day-ahead-real-time price spreads while capturing a similar amount of time as the as the high net load, cold weather, and high renewable generation drivers.

The last paragraph on page 6 is hard to follow (e.g. "...are an increase 64%...") It is stated that 74% of peak value hours overlap with a detected event, but in the paragraph before it states that the events occur in more than 75% of peak hours.

Thank you for highlighting the need for more clarity in this important section. First, let us clarify the 74% and 75% numbers the reviewer specifically asked about:

- "Unforeseen intraday variance, high net load, cold weather, or high renewable power generation conditions are present in over 75% of the hours with peak transmission value."
 - $$\frac{\# \text{ hours with unforeseen intraday variance, high net load, cold weather or high renewable generation}}{0.05 * \# \text{ hour studied}} = 0.75$$
- "74% of the value resulting from the peak value hours overlaps with a detected unforeseen event"
 - $$\frac{\text{Transmission value in hours with unforeseen intraday variance}}{\text{Transmission value during top 5% of hou}} = 0.74$$

Our approach to clarifying this paragraph was primarily to draw connections to Figure 5a so that readers can visualize the numbers being shared. We also made other small edits to improve the readability.

The font in figure 5 is too small and not really possible to read.

Figure 5 has been edited to have larger font sizes.

The authors point out, correctly, the marginal nature of the analysis since it is based on historical marginal prices. The investigation of market depth and saturation is therefore an important addition. However, to what extent has the approach been validated? Did you attempt to measure the accuracy of the approach? Impacts on transmission expansion on prices is likely to be non-linear. Would non-linear impacts be captures? This should be elaborated in the paper. A brief description of the approach within the main text would be helpful for readability.

Thank you for identifying that our approach to investigating market depth and saturation requires elaboration in general and specifically within the main text. First, we will address your questions and then discuss how we have edited the paper in response.

- *Would non-linear impacts be captured?* Yes, we used cubic functions in order to capture non-linear impacts, including the steep growth in a supply curve that is often seen at high net-load levels (i.e., the "hockey stick" shape).
- *Did you attempt to measure the accuracy of the approach?* We did measure the accuracy of the supply curve models that underpin the approach and excluded models with poor accuracy. Specifically, we exclude models from the analysis if less than two-thirds of the hours during the corresponding bi-monthly period are not within a specified prediction accuracy tolerance level. For the 5256 node-period supply curve models computed for the comparison of transmission value to cost, 99.8% passed the screening and at most 2 failed per node. More details are

available in the third paragraph of the “Transmission market value accounting for market depth” methods section.

The approach then assumes that an increase in power flow between two regions has the effect of increasing net load in the exporting region and decreasing net load in the importing region by the same amount and that net load shifts due to imports/exports have the same effect on prices as net load shifts due to other drivers. This is based on first principles (assuming no transmission losses), so we did not attempt to measure the accuracy of this aspect of the approach.

- *To what extent has the approach been validated?* The use of statistically based rolling cubic supply curves for ~2-week periods was previously validated in reference [47]. Our “steepest slope method” for obtaining lower value estimates – applying the most extreme supply curve gradient observed for a given node over the entire study horizon to 1) periods with models that were excluded and 2) hours with a large residual (i.e., the *(net load, price)* pair is far from the supply curve model) – has not been otherwise validated. It assumes prices will change quickly with small changes in net load and applies a location-specific definition of “quickly.” We developed this approach in response to a lack of literature about how sensitive price spikes and dips are to changes in net load when net load is not their only major driver.

Paper edits: We have added a one-paragraph summary of the approach in the main body (2nd paragraph of Market depth and saturation effects section) along with a figure showing two examples of the approach to make clear that non-linear impacts would be captured.

In the first paragraph of the conclusion, what is meant with “frequent repricing instead of continuous repricing”? Also, the classical explanation of the missing money problem is price caps and possible other regulatory interventions that prevent the price to reach sufficiently high levels during supply scarcity.

Similar to price caps, other simplifications and interventions in real-world pricing procedures suppress price volatility. One such simplification is that we reprice the system frequently (e.g., every 5 minutes) instead of repricing in continuous time, which is the theoretical ideal. This simplification is what is meant by “frequent repricing instead of continuous repricing.” However, after editing the discussion & conclusions section, this text has been removed.

The authors state that the number of merchant lines built has been small in the US. Could you be more specific? How many? The authors allude to non-economic constraints as an explanation. What do you have in mind? Could it be that transmission value, the way it is calculated in the paper, is not a good proxy for investors (e.g. due to lumpiness of investments)?

This is an important comment, but a difficult one to address. No compiled data source exists across the U.S. to provide a precise statement about the total number of merchant lines in operation. However in PJM, the largest electricity market region in the U.S., less than 1 percent of its total transmission capacity is made up of merchant lines. We have added this sentence and corresponding citation into the discussion section of the report.

In relation to the reviewer's comment on non-economic constraints, what we had in mind were the transaction costs and institutional challenges involved with capturing the transmission value that we calculate in the paper. Merchant lines are developed by independent entities who must find customers (typically load serving entities) that buy or subscribe to offtake capacity on the new transmission lines. However, those customers are oftentimes utilities that own and operate their own transmission lines. These complex competitive dynamics are one key reason why it is difficult to capture the full market value via a merchant option. We have added a short sentence offering this explanation in the paper.

Have the authors compared their results to transmission value estimates from model-based studies? That would be a good extension that would link the analysis to other studies out there, and also add to validation of results.

We agree that comparison of these results to modeling studies would be a great extension to this work, and in fact, are already working on a large project with this as its goal. We have done some preliminary comparison to existing studies, and comparing on an apples-to-apples basis is quite challenging and beyond the scope of the current manuscript. To share some of our first impressions, we have observed some important differences between modeled and our price-based analysis. Our most consistent *preliminary* finding across different models was that modeled transmission value is less concentrated in time than our estimates based on market prices. That said, this work remains preliminary and thus we can only recommend this area as a prime target for future research.

In methods, the authors claim that LMP markets are the only ones signaling the value of transmission. What about zonal markets?

The reviewer is correct that zonal markets would provide signals of the value of transmission between zones, and we have updated the paper accordingly.

The authors should elaborate on the implications of the study. To what extent should these types of analysis be used to inform transmission planning (across the various types, including merchant)? Inform policy? Other uses?

Thanks for this feedback—we agree. We have added a section, “Implications for modelers, investors, planners, and policymakers” at the end of the paper that answers these questions.

Reviewer #3

The one section that I had trouble following is the section accounting for “market depth.” Since this is a very important and complicated section the methods should be clear at least conceptually. More conceptual presentation should be in the text and not just in the methods section since this analysis is very important. [...] I understand the “market depth” analysis on a conceptual level, but “market depth” is not enough. We need supply and demand functions for each relevant market area and the entire ISO is unlikely to be a relevant market area since there is internal network congestion that yields smaller market areas. Anyway, the presentation in this section needs to be expanded and not rely only on the methods section. The issue can be presented with a couple of figures and then the key features of the analytical approach discussed in the text. The reader can then be sent to the methods section for even more details regarding each step.

Thank you for identifying that our approach to investigating market depth requires a clearer and more conceptual presentation within the main text. We have added a one-paragraph summary of the methods in the main body (2nd paragraph of Market depth and saturation effects section) along with a figure, as you suggested, showing two examples of the approach with differing outcomes. The figures show that we create location-specific supply functions and use them to determine how much of the additional capacity would be utilized and how quickly prices converge.

To the comment about demand functions, any demand responsiveness is baked into the supply function, because it is based on actual historical load and price observations. So, we treat demand as perfectly inelastic. We added the following text to the methods section to make this explicit: “Demand is modeled as perfectly inelastic, because demand responsiveness is already reflected in the observations and therefore captured as a supply resource in the fitted supply curves.” Previous analyses, including the following sources, have also shown short-run electricity demand in the United States to be highly inelastic:

1. Burke, P. J., & Abayasekara, A. (2018). The price elasticity of electricity demand in the United States: A three-dimensional analysis. *The Energy Journal*
2. Arent, D. (2006). Regional Differences in the Price-Elasticity of Demand for Energy. <https://www.nrel.gov/docs/fy06osti/39512.pdf>

Finally, we discuss the important and complex consideration of market size. In the context of interface pricing for interregional trade, [3] determines that (at least for MISO, SPP, and PJM) the marginal generators that support imports and exports are distributed throughout the ISO, not just specific buses near the borders, suggesting that the entire ISO is a relevant market area. However, we are building separate supply curves for different pricing nodes within each region because their prices can differ due to internal congestion (as the reviewer points out), suggesting that the zone is a relevant market area. In reality, prices have a complex relationship with demand in many market areas: net load in their zone, net

load in neighboring zones (within and outside their ISO), total net load in their ISO, and total net load in the entire interconnected system. We are interested in exploring these dynamics in future work, but felt developing such a multivariate modeling methodology was outside the scope of these revisions. Instead, we have provided a more aggressive test of market depth that uses 50% smaller market areas as a sensitivity analysis of this important consideration. The sensitivity analysis is referenced in the paper and the results (copied below) are included in the supplemental information.

3. Potomac Economics (2020). OMS-RSC Seams Study: Interface Pricing.

https://www.spp.org/documents/62738/seams%20study_miso%20imm_interface%20pricing%20study_final.pdf

Range of transmission market value estimates accounting for market depth of a 1 GW transfer capacity increase, when assuming the relevant market size is 50% of the size used in the main paper, relative to marginal transmission value (real-time). Grey and white bands are used to improve readability; they do not communicate information about the results.

Comparison of project costs to market value estimates that account for market depth assuming the relevant market size is 50% of the size used in the main paper. (a) Value-to-cost ratios where the value is the midpoint of the value estimate range in (b) and the cost estimate assumes 60-year depreciation. The horizontal dashed line at 1 represents the “break-even” ratio. (b) Range of market value (real-time) estimates accounting for market depth and annualized project cost estimates.

A very basic question is “Why does anyone care about answering the question posed above?” The past is the past. Investments take place in the future. We anticipate the generation mix and the distribution of nodal prices will change dramatically in the future as the types and location of generating facilities, storage facilities, and demand change in the future (EVs, heat pumps, data centers, etc.). The transmission facilities necessary to support this generation, storage and demand configuration are likely to be quite different from those made in the past or could have been made hypothetically to support historical generation and transmission configurations. This is why models are used to examine future scenarios and their attributes. As the authors note, the answer is that the value of the historical analysis is to demonstrate that barriers other than “economics” have kept socially valuable transmission links from being built in the past. I agree with this conclusion. However, the paper would have been much better if it had elaborated on what those barriers are likely to have been since these barriers likely continue to exist today and will create problems going forward as they have in the past. Relying on a very limited discussion of references is not enough.

Thank you for this thoughtful comment. We agree with this comment that one key contribution of the paper is to answer, “What would the value of this set of transmission links have been over the 2012-2022 period given the portfolio of generating plants in place during this period if the transmission links, one at a time, had been built prior to this period at the construction costs estimated in the paper?” The follow up point from the reviewer is, why do we care about this past value, when transmission decisions need to be built on assessments of future value vs. cost? The reviewer indicates that our answer to this question, that this analysis shows that non-market barriers have prevented optimal transmission expansion, is a useful conclusion, but needs more discussion. In particular, the reviewer suggests expanding our discussion of existing barriers to transmission expansion, as these barriers are likely to continue to limit optimal transmission expansion into the future.

In response, we first would like to highlight a larger set of answers to the ‘why do we care about past value’ question. The new subsection of the Discussion and Conclusion section, “Implications for investors, modelers, planners, and policymakers,” attempts to explicitly address the why-do-we-care question. We discuss how this work could be used to refine forward looking modeling practices so models more realistically simulate transmission value, how the work can be used to address concerns about cost-allocation between regions, and how the work can inform specific regulatory processes from FERC. We admit that each of those points could also be framed as addressing a type of barrier, but we think, for example, improving modeling practices is topically quite different from a logistical-type barrier such as the need to coordinate large and complicated planning processes across multiple regions and many stakeholders to ensure optimal interregional transmission.

Second, we have refined the earlier portion of the Discussion and Conclusion section to more comprehensively discuss barriers – in this section we have added a full paragraph discussing specific barriers from multiple citations. We have also further linked the discussion of the drivers of observed market transmission value to potential efforts to improve modeling of transmission

benefits. Of course, challenges to accurately and comprehensively modeling transmission benefits are themselves a barrier.

I could not figure out whether the paper uses day-ahead or real time nodal prices to value transmission investments. Day-ahead prices are hourly forward financial commitments by generators and load to sell or buy specific quantities at the hourly market-clearing prices. (While these are forward financial commitments there are also physical implications, including unit commitment, max and min generation, congestion contingent bids, etc. A surprisingly large fraction of generators in the day-ahead market self-schedule rather than bid and are price-takers.) The real time market supports the physical dispatch of generators, storage, and price-sensitive demand to balance supply and demand and meet operating reliability criteria. The real time dispatch may vary from day-ahead schedules because demand is higher or lower than forecast, generating plants have forced outages, more cheap imported energy may become available, etc. For example, upward deviations in generation from the day-ahead schedule and day-ahead price to meet dispatch instructions are compensated at the real time price for the increment in generation in excess of the day-ahead schedule only. Most generation and demand settles at the day-ahead price and from the perspective of the buyer and the seller at different nodes, the difference in the day-ahead prices between those nodes is the value of a (marginal) increase in transmission capacity between the two nodes. However, it appears that the paper uses real time prices to value transmission. It partially relies on a paper in a footnote whose analysis is not described. This section needs to be clarified and a simple couple of examples presented to justify using the real time prices if that's what they have done. In addition, it would be useful to present valuations using both day-ahead and real-time prices.

Thank you for identifying that the results lacked clarity as to which market prices were used and for your insights into the differences between day-ahead and real-time markets. On the clarity point, we have now ensured that all figure captions clearly state which market prices were used. Additionally, we have added language to the first paragraph of the first results section ("Geospatial patterns of transmission value") stating that transmission market value is market-specific and that the real-time market is the main focus of this paper. Previously, this was stated in the methods section, which does not appear until after the results which likely contributed to the confusion.

On the broader point of day-ahead and real-time market signals of transmission value, we agree that each can be relevant, depending on the questions you want to answer. To make the market differences more apparent to readers, we have expanded our paragraph on page 5 that compares day-ahead and real-time market prices and transmission value into its own subsection and introduced a figure visualizing the differences. Readers who have a preference for day-ahead transmission value can use the value ratios in the left panel of this figure as a rule-of-thumb for adjusting real-time values presented elsewhere. Additionally, we have added day-ahead versions of Figures 2 and 3 to the supplemental information. Ultimately, our decision to use real-time transmission value as our default comes down to three factors:

1. We are most interested in transmission's ultimate value to the system and real-time prices reflect physically binding dispatch decisions under the actual operating conditions of the system.
2. Past work (the source below, in particular) has identified real-time uncertainties as a critical driver of the benefit of geographic diversification that transmission provides. Given this literature, which is based on a modeling framework, we believe it is important to use prices that reflect real-time uncertainties in our empirical assessment of transmission value.
 - a) K. Van Horn, J. Pfeifenberger, and P. Ruiz, "The value of diversifying uncertain renewable generation through the transmission system," Sep. 2020, [Online]. Available: <https://hdl.handle.net/2144/41451>
3. The Western Energy Imbalance Market (WEIM) is a real-time only market. So, if we focused on the day-ahead timeframe, we would need to either exclude these important links from the analysis altogether or use day-ahead prices for some links and real-time for the WEIM links, and we believe it is important to avoid "mixing apples and oranges" in this way as much as possible.

Unfortunately, we were not able to locate the footnote you refer to in order to directly address that portion of your comment, but we hope that the clarifications above and in the paper have addressed the ambiguity around this topic that existed in the original draft.

The authors have put together a very useful data set on transmission investments costs. However, two issues arise. First, the paper should demonstrate that the transmission cost data that are being used are representative of the "new" links being analyzed. The cost of a 20 mile upgrade of a 250kv line on an existing right of way may not be representative of a new 100 mile 500kv line on a new right of way. Second, a new link with a large transfer capacity is highly likely to require additional reinforcements at one or both ends of the AC system. For example, when the HVDC link from Quebec to Boston was built in the late 1980s, rather substantial investments in the New England AC network (NEPOOL at that time) were required to accept and disperse the additional energy injected at a new location outside Boston (an investment in PJM was also required for reliability). Failure to include these costs underestimates the true costs of the link.

Thanks for the positive feedback and the important comments about how to better describe the robustness of the transmission cost estimates we compiled. We completely agree that there can be a significant cost difference between a 20-mile upgrade of a 250 kV line and a 100-mile 500 kV line. Our method of collecting and comparing costs focused on identifying actual completed or proposed transmission lines which we could confidently claim were establishing or already established an electrical link between the regions we analyze for transmission value. The cost data we collected are public, and we have added a table into the methods section of the paper to provide more transparency around where these data were sourced, and which explicit online or proposed transmission project cost estimate we used in coordination with our transmission value links. The cost estimates are the all-in costs to develop and construct the new transmission lines, thus the costs incorporate the costs to interconnect the transmission

line into the current transmission system (i.e. inclusive of investments the project must make to develop the broader transmission network, if any). The projects we relied upon were multi-million and oftentimes billion-dollar investments to achieve the transfer capability indicated. The table we added to the methods is provided below for convenience.

Project Name	Link Name	Line Capacity (MW)	Technology	Geographic Scope	Status	Business Model	Total Cost (\$2022 Million)	Annualized Cost (\$2022 million/yr)	Source
Tehachapi Renewable Transmission Project	CAISO_LosAngeles <-> CAISO_DesertHub	4500	AC	Regional	Complete	Utility	\$3,765	\$180	Gorman (2019)
Sunrise Powerlink	CAISO_LosAngeles <-> CAISO_DesertHub	1000	AC	Regional	Complete	Utility	\$2,371	\$114	Gorman (2019)
Ten West	CAISO_DesertHub <-> EIM_AZ	2700	AC	Interregional	Non-Complete	Utility	\$302	\$14	Grid Strategies (2023)
Grand Prairie Gateway	MISO_WI <-> PJM_IL	1000	AC	Interregional	Complete	Utility	\$326	\$16	Gorman (2019)
Eldorado-Ivanpah	CAISO_DesertHub <-> EIM_AZ	1400	AC	Interregional	Complete	Utility	\$425	\$20	Gorman (2019)
Plains & Eastern	MISO_LA <-> SPP_SouthHub	4000	HVDC	Interregional	Non-Complete	Merchant	\$2,880	\$138	Gorman (2019)
Devers - Valley No. 2 Transmission Project DPV2	CAISO_DesertHub <-> EIM_AZ	1250	AC	Interregional	Complete	Utility	\$970	\$47	Gorman (2019)
Transwest Express	CAISO_DesertHub <-> EIM_UT	3000	HVDC	Interregional	Non-Complete	Utility	\$3,456	\$166	Grid Strategies (2023)
Pecos West Intertie	EIM_AZ <-> ERCOT_WestHub	1500	HVDC	Cross-Interconnect	Non-Complete	Merchant	\$1,500	\$72	Online Webpage (2023)
North Plains connector	MISO_ND <-> EIM_UT	3000	HVDC	Cross-Interconnect	Non-Complete	Merchant	\$3,200	\$153	Online Webpage (2023)
MISO Multi-Value Projects #3	MISO_INHub <-> MISO_MIHub	1486	AC	Regional	Complete	Utility	\$1,358	\$65	Gorman (2019)
MISO Multi-Value Projects #2	MISO_MNHub <-> MISO_INHub	3377	AC	Regional	Complete	Utility	\$2,143	\$103	Gorman (2019)
Southern Cross (Spirit)	MISO_LA <-> ERCOT_SouthHub	3000	HVDC	Cross-Interconnect	Non-Complete	Utility	\$2,700	\$129	Grid Strategies (2023)
Clean Path New York	NYISO_NYC <-> NYISO_NorthHub	3800	HVDC	Regional	Non-Complete	Merchant +State	\$3,780	\$181	Grid Strategies (2023)
Empire State Connector	NYISO_NYC <-> NYISO_NorthHub	1250	HVDC	Regional	Non-Complete	Merchant	\$1,765	\$85	Grid Strategies (2023)
CREZ	ERCOT_WestHub <-> ERCOT_SouthHub	18500	AC	Regional	Complete	Utility	\$8,119	\$389	Gorman (2019)
Southwest Minnesota Wind Expansion	MISO_SD <-> MISO_MNHub	800	AC	Regional	Complete	Utility	\$395	\$19	Gorman (2019)
MISO Multi-Value Projects #1	MISO_WI <-> MISO_MNHub	9137	AC	Regional	Complete	Utility	\$3,596	\$172	Gorman (2019)
Grain Belt Express	SPP_SouthHub <-> MISO_INHub	5000	HVDC	Interregional	Non-Complete	Merchant	\$8,065	\$387	Gorman (2019)
Gateway West	EIM_BPAHub <-> EIM_UT	3000	AC	Interregional	Non-Complete	Utility	\$4,707	\$226	Grid Strategies (2023)
Wyoming Intertie	EIM_UT <-> SPP_WestHub	3000	HVDC	Cross-Interconnect	Non-Complete	Merchant	\$1,000	\$48	Online Webpage (2023)
SOO Green	PJM_IL <-> MISO_MNHub	2100	HVDC	Interregional	Non-Complete	Merchant	\$4,000	\$192	Grid Strategies (2023)
Gateway South	EIM_AZ <-> EIM_UT	3000	AC	Interregional	Non-Complete	Utility	\$4,707	\$226	Gorman (2019)
SPP Priority #1	SPP_SouthHub <-> SPP_WestHub	2625	AC	Regional	Complete	Utility	\$1,059	\$51	Gorman (2019)
SPP Priority #2	SPP_KS <-> SPP_NorthHub	375	AC	Regional	Complete	Utility	\$376	\$18	Gorman (2019)
Three corners Connector	SPP_SouthHub <-> EIM_AZ	3000	HVDC	Cross-Interconnect	Non-Complete	Merchant	\$2,000	\$96	Online Webpage (2023)

I turn next to AC network issues. This paper is a transmission link by transmission link analysis. But AC systems obey Kirchhoff's laws. Accordingly, a large new transmission link can affect all of the nodal prices on a transmission network. This also has implications for AC network upgrades as I pointed out earlier. At the very least, the paper should at least mention this simplification and make the case that it is not important for the analysis.

The reviewer is correct that the original draft only briefly mentioned AC network issues when it said, "The line may not be fully utilized even when there is congestion due to power flow and security constraints." Based on your feedback, we have made this important point much more robust:

"The line may not be fully utilized even when there is congestion due to security constraints and power flow dynamics. The latter refers to the fact that the path of power flow on alternating current (AC) networks cannot be arbitrarily prescribed, because the electricity flows across transmission facilities according to Kirchhoff's Laws based on the power consumption of loads and the voltage magnitude and real power injection established by generators. While direct current (DC) lines offer greater control over power flows, the surrounding AC system dynamics could still limit utilization."

Separately, when addressing the price impacts of transmission expansion, we clarify that this is a link-by-link analysis, as the reviewer states, and that we are not considering how prices and flows could change in tertiary regions or zones in response. The clarification is included both in the "Market depth and saturation effects" results section and in the methods subsection "Transmission market value accounting for market depth," which now states:

"In this paper, we estimate the effect of such price convergence and the depth of markets on transmission value for each link individually. That is, we only consider the change in prices for two nodes at a time, and we assume each link is the only transfer capacity addition when analyzing it. We do not consider a case where the capacities of all, or a group of several, analyzed links are expanded simultaneously."

If one wanted to assess the price impacts of the joint expansion of all or several links in the paper, it would be important to develop additional methods to capture interdependencies.

I could not figure out where day-ahead and real time nodal prices came from for the Southeast and Northwest since they do not have RTO/ISOs. Please make sure that the answer is clear.

We thank the reviewer for pointing out that our selection of pricing nodes in these regions is not clear. The answer is that, in the West, prices come from the Western Energy Imbalance Market. In the Southeast we use interface and external pricing nodes that MISO and PJM maintain for TVA and Duke, respectively. We have updated the scope section where we discuss pricing nodes as follows:

"...Each of the remaining nodes is either a load, generator, tie generator, interface, or external node. The non-ISO Southeast nodes are of the latter two types as reported by MISO and PJM. Prices in the non-ISO West are Western Energy Imbalance Market prices reported by CAISO."

My suggestion is to consider comparing your results to the preliminary DOE analysis of National Interest Transmission Corridors (<https://www.energy.gov/gdo/national-interest-electric-transmission-corridor-designation-process>.) and the DOE/NREL analyses of future transmission needs.

This is a good suggestion. We have added discussion of the NEITC (and thus also the DOE Needs study) into the new 'Implications for investors, modelers, planners, and policymakers' subsection of the Discussion/Conclusion. The key point being that the NEITC provides an example of taking both forward looking and historical pricing and congestion patterns into account when planning for transmission expansion. We also note that the selection of corridors largely matches the key findings from our work, in that they are located in high value (based on historical pricing analysis) links between regions and interconnections.

Review responses, round 2: Electric transmission value and its drivers in United States power markets

Summary note to all reviewers

We appreciate the continued engagement with our work from both reviewers. It appears that our first round of revisions succeeded in addressing most of the reviewers' comments. The remaining comments have been addressed in the paper, which improved the clarity of several key figures and the transmission cost inputs. Please see point-by-point responses to the latest comments below.

Reviewer #2

Figure 4 is a nice addition, but could benefit from a more extensive description in the figure text. For instance, what does links and hours refer to in the middle panel?

The middle panel shows time series at an annual resolution for two quantities:

1. Percent of links where transmission value is greater in the RT market than in the DA market
2. Percent of hours where transmission value is greater in the RT market than in the DA market

The first sums across all hours in the year for each link to determine which market has greater transmission value and then reports the percentage of links where it was the RT market with greater value. The second does not aggregate by link and instead counts the number of hours across all links where the RT value was greater than the DA, and reports that count total as the share of all hours.

We've updated the figure caption to be more explanatory, as suggested:

Comparison of transmission market value in day-ahead and real-time markets. Excludes links connected to the non-ISO West where there is not a day-ahead market. The set of links in the early years is smaller due to data constraints. Left: Average in the real-time market relative to average in the day-ahead market for 1) transmission market value and 2) wholesale electricity prices. Center: Prevalence of greater real-time transmission value in terms of 1) the share of links where annual real-time value was greater than day-ahead and 2) the share of hours across all links in which real-time value was greater than day-ahead. Right: Magnitude of the average hourly difference between day-ahead and real-time transmission value presented separately for hours in which 1) real-time value was greater than day-ahead value and 2) day-ahead value was greater than real-time value.

I still find the section on conditions during times of peak transmission value a bit hard to follow and suggest you make one more attempt to clarify the discussion around Figure 6. In particular, why are the identified events (high net load, cold weather, high renewable generation) listed as separate factors as well as parts of the unforeseen events? I believe the reason is that they contribute to transmission value during classified unforeseen events and at other times, but this is not straightforward from the discussion. Why is cold weather written within the yellow bar for high net load? Does it occur at the same time as high load? If so, should high load be identified in the red bar for cold weather? In figure 6b, would it make sense to also display the calculated transmission value?

First, let us clarify the methods. Then, we will explain the components of Figure 6 and how we've updated the paper to make it more clear. For each of the four conditions (unforeseen intraday variance, high net load, cold weather, high renewable generation) we study, we independently determine which link-hours have the condition. So, a link-hour may have 0, 1, 2, 3, or all 4 conditions. In Figure 6a, a given link-hour should only be counted once toward explaining the value in the top 5% of hours, regardless of how many conditions it has. For link-hours with multiple conditions, the associated value could be assigned to different columns of the figure. We intended for the figure to be read from left to right and we include an hour's value in the first (i.e., left-most) valid column:

- The Unforeseen Intraday Variance column includes all hours with this condition. We distinguish between hours with *only* this condition (darkest blue) and hours with other conditions too (lighter blue labeled segments).
- Then the High Net Load column considers those hours that are not already represented in the Unforeseen Intraday Variance column (to avoid double counting) and have high net load. Again, we distinguish between the value with only high net load and that with overlap.
- Then the Cold Weather column considers those hours that are not already represented in the Unforeseen Intraday Variance or High Net Load columns that have cold weather. In theory there could be overlap with the remaining condition (high renewable generation) but in practice this overlap is negligible.
- Finally, the only hours with a condition remaining are hours with high renewable generation and no other conditions. (The right-most condition column will never have overlapping conditions, regardless of which order one chose to put the conditions in.)

To make all of this clearer to the reader we have updated Figure 6a and the associated text:

Figure 6 Relationships between four key conditions and peak transmission value (real-time market). Aggregate results for all 52 studied links within or between ISO or RTO regions, excluding those in the non-ISO West and Southeast. (a) Contribution of key system conditions to transmission market value during peak value hours (top 5%). This figure should be read from left to right. When multiple conditions are present at the same time, the associated value appears in the first (i.e., leftmost) applicable column and is identified by lighter segments and “& [overlapping condition]” labels. (b)...

Regarding displaying the calculated transmission value in Figure 6b, we explored some options for annotating the figure with that information. Ultimately, we didn't succeed in finding an option that didn't result in too much information for the small space.

Figure 8 is a nice addition that makes it easier to understand the analysis around market depth. The sensitivity case with a 50% reduction in market size is also a good addition. The analysis still considers a 1GW expansion only. The new table 2 indicates that most transmission lines have a larger capacity. What would happen for a larger expansion of capacity, say 2GW or 5GW? Would Figure 9 look different in that case? Would the effect be similar to the case assuming reduced market size?

We're glad that you found the additions to the market depth analysis effective. Regarding larger capacity expansions, let's first consider the CAISO South <> Arizona link where the paper analyzes expansions of 1000 MW (for Figure 9), 1250 MW, 1400 MW, and 2700 MW (for Figure 10) at both the original and 50% market sizes. The impact on value of these different cases is summarized in the table below. For this link, reducing the market size by 50% decreases the high and low ends of the per-capacity value range by 10-15% and 20-30%, respectively, depending on the expansion capacity (right-most table columns). Meanwhile, an expansion of 2700 MW relative to an expansion of 1000 MW decreases the high end of the per-capacity value range by 15% (original market size) or 20% (50% market size) and decreases the low end of the per-capacity value range by 30% (original market size) or ~40% (50% market size). So, in this case I would say that moving from 1000 to 2700 MW has a similar effect on the per-capacity value of a 50% reduction in market

size for a 2700 MW expansion, but a greater effect than a 50% reduction in market size for a 1000 MW expansion.

	Low end of value range (\$/MW-year) as a percentage of low end of value range for 1000 MW expansion		High end of value range (\$/MW-year) as a percentage of high end of value range for 1000 MW expansion		Value (\$/MW-year) for 50% market size as a percentage of value for original market size	
	Original market size	50% market size	Original market size	50% market size	Low end of value range	High end of value range
1000 MW	-	-	-	-	80%	90%
1250 MW	94%	91%	97%	96%	78%	89%
1400 MW	91%	86%	96%	94%	76%	88%
2700 MW	70%	61%	85%	80%	69%	85%

Another example you can look at to see the effect of larger capacity expansions is the NYISO North <> South cases with 1250 and 3800 MW expansions in Figure 10.

Ultimately, there are many different combinations of test cases one can consider here, and we have tried to make the reader generally aware of these sensitivities. To the reviewer, we would like to emphasize that, when comparing value to cost in Figure 10 and supplemental Figure 17 (50% market size sensitivity), we use the stated expansion size in the analysis and do not simply assume that the results in Figure 9 generalize to any expansion size.

Small detail: The end of the discussion/conclusion now uses the term cost assignment. I believe cost allocation is commonly used for this purpose.

We have updated our terminology from assignment to allocation, as recommended.

The new section on implications is a good addition to the manuscript

Thank you for this feedback!

Reviewer #3

The responses to the referees' report are more than satisfactory. One item needs a bit more discussion. This is the table of projects that appears on page 15 of the response to referees and in the methods section. The body of the paper should make it clearer that more than half of these projects have not been completed so the costs are estimated costs. I have never seen a major transmission project that is completed at or below the estimated costs. So, I think that the actual costs are likely to be higher. (Some of these projects have faced permitting problems and may not even be built.)

We have revised the paper to make it more clear which costs are for projects that have not been completed by differentiating the costs for complete and not complete projects in Figure 10b. (Previously this distinction was only made in Figure 10a.)

Also, the cost estimates are in different year dollars since the construction schedules are different.

We had made an effort to control for the issue of different year dollars by converting all cost estimates into 2022 dollars based on the year that the cost estimates were made. However, we did not have access to a full financial model, which would reflect construction schedules, for each of the projects identified. We are not aware of a source of this information, otherwise we would gladly incorporate it into the paper's cost estimates. We have added the following text to the "Transmission cost estimates" section to make it clear to the reader that there remains uncertainty and variability in these cost estimates, as you point out:

There is uncertainty in the transmission costs identified here, especially for projects that have only been proposed and not completed, as they may incur cost overruns, not represent the full all-in project cost, or not reflect differences in construction schedules.

I think that this table should be in the body of the paper and not buried in the methods section, which almost nobody reads. This will make these attributes of the cost data clearer to readers.

We continue to believe that the methods section is the appropriate place for the detailed project-level information found in Table 2. However, we agree with the reviewer that this table should be referenced in the body of the paper so that readers who are otherwise not interested in the methods are aware of it. We have added two such references: One in the text of the "Transmission costs compared to market value" section, one in the caption of Figure 10.

Finally, I am familiar with a few of these projects and, contrary to the statement in the response, they do not include estimates (or actual) of the full interconnection costs.

As mentioned above, we have added the following text to the "Transmission cost estimates" section to make it clear to the reader that there remains uncertainty and variability in these cost estimates, as you point out:

There is uncertainty in the transmission costs identified here, especially for projects that have only been proposed and not completed, as they may incur cost overruns, not represent the full all-in project cost, or not reflect differences in construction schedules.

Review responses: Electric transmission value and its drivers in United States power markets

Reviewer #4

The basic methodology is to focus on locational prices, which include congestion costs, and to interpret the difference in prices between two locations as an estimate of the marginal value of transmission. This is true as a marginal value of energy transmission defined as injection at one location and withdrawal at another. However, this is not necessarily the same thing as the value of transmission line expansion. The difficulty centers on two choices.

We agree with the reviewer that transmission's marginal market value (defined as the price difference between two locations, as stated by the reviewer) does not fully capture all benefits that transmission expansion may provide. In subsequent sections we will address the two specific choices highlighted by the reviewer. First, we want to highlight an existing passage of the paper where we echo this high-level point made by the reviewer. On page 3, the following text states that transmission benefits are reflected in market values of transmission to varying degrees:

There are several categories of societal benefits that transmission may provide, including resource adequacy, resilience, risk mitigation, and reductions in emissions, market power, capital costs and production costs [21]. These concepts are priced into wholesale electricity markets to varying degrees, and similarly reflected in market values of transmission. Centralized system planners, however, separately quantify each benefit they consider and typically use production cost savings as the main economic benefit. We do not offer a one-to-one comparison between transmission's market value and each of its modeled benefit types, instead using market value as an aggregate signal. Empirical prices precisely reflect actual system conditions and market participant behavior as they occurred in the past, providing insights that cannot be gained from system models.

First, "The transmission value analyzed in this section is a pairwise quantity between two wholesale market pricing nodes (i.e., a "link") defined as the mean absolute locational marginal price difference between the nodes over time (in units \$/MWh)." This use of the term "link" could easily be misinterpreted to imply that there is a transmission line connecting the two locations and that the price difference between two locations would be caused by constraints on that line.

We appreciate the reviewer identifying this risk of misinterpretation of the term "link." We have made the following changes to the paper to mitigate this risk:

- Removed this term from the introduction
- Added the following sentence to the first paragraph of the "Geospatial patterns of transmission value" section, where the term "link" is introduced: A "link" is uniquely

defined by two pricing nodes; it does not correspond to a specific transmission line, and there may be zero, one or multiple existing transmission paths between a link's nodes.

- Added the following sentence to the Figure 1 caption: The line segments depict which pairs of wholesale market pricing nodes are analyzed and do not portray existing transmission lines.

But any “line may not be fully utilized even when there is congestion due to security constraints and power flow dynamics. The latter refers to the fact that the path of power flow on alternating current (AC) networks cannot be arbitrarily prescribed, because the electricity flows across transmission facilities according to Kirchoff's Laws based on the power consumption of loads and the voltage magnitude and real power injection established by generators.” In other words, the price difference may be, and typically is, caused by simultaneous limits on many transmission lines, including contingency security constraints.

We agree with the reviewer's comments about the effect of network constraints on price differences. To clarify this within the paper, we added the following text to the section “Methodology foundation: market price signals”:

A price difference between two nodes could be caused by congestion exclusively on a line directly connecting them, but it typically is the result of simultaneous limits on multiple pieces of transmission infrastructure.

Furthermore, transmission line expansion does not typically follow a constant-returns to scale cost structure. Hence the ex ante marginal conditions may not be an accurate indicator of beneficial transmission investment.

We understand this comment to be focused on the “Market depth and saturation effects” and “Transmission costs compared to market value” sections. This is where we consider ex ante marginal conditions and demonstrate the non-constant returns to scale mentioned by the reviewer. Specifically, Figure 8 provides two illustrations of the declining returns that come with increasing transfer capacity.

The perceived accuracy of the transmission value accounting for market depth estimates we derive depends on one's level of confidence in the methods and assumptions. To support our methods, we provide a detailed description in the Methods section, a summary and two examples in the main text, algorithmic pseudocode in the supplemental materials, and sensitivity analysis (i.e., higher and lower estimates, multiple market size definitions). A key assumption employed here is that the market is static (page 11). That is, it does not account for generator entry and exit, demand changes, shifts in market power, or other dynamic effects. The extent to which this assumption would have been violated in practice certainly affects the accuracy of the results. Relaxing this assumption would require a multi-agent framework, which was beyond the scope of this analysis but could be an interesting area of future research.

In the “Market depth and saturation effects” section, we added additional language to clarify that the value accounting for market depth is an estimate, not a definitive quantity like the marginal transmission values analyzed earlier:

Unlike the directly observable marginal transmission value metric, this value is an estimate for a hypothetical situation.

Reviewing the language in the “Transmission costs compared to market value” section (copied below), we believe it is clear in that section that these results alone should not be used to determine whether any individual transmission investment is beneficial.

To better contextualize our market value estimates, we compiled data on the costs to construct individual transmission projects across different regions of the United States. [...] Such value-to-cost comparisons should not be used to assess the full cost and value of any individual, specific transmission investment. However, the value-to-cost ratios we present in Figure 10 do provide some indication of the net economic value of transmission development, focusing here solely on energy market value and excluding other possible benefits of transmission investment.

In other words, material congestion costs are a necessary but not sufficient indicator of beneficial transmission investments.

We also want to highlight that, beyond congestion costs, transmission investments can be considered beneficial because they support system reliability or public policy directives, which are outside the focus of this paper.

This seems to take us back to the use of models comparing existing operations against a new profile created by a candidate transmission investment.

We agree with the reviewer that models are an important tool for evaluating candidate transmission investments. Here is a collection of statements from the paper describing how we view this work as complementary to modeling:

- This paper contributes to a more complete understanding of transmission’s value, cost-effectiveness, and market barriers and aids efforts to improve modeling practices by identifying key patterns and value drivers for use in model validation [page 3].
- A weakness of such models is that often they do not replicate real-world conditions that affect the benefits of transmission, conditions such as forecast errors, extreme weather, and infrastructure outages [p.2].
- While our approach is not meant to replace models, it is intended to help identify key mechanisms that may lead to biased model estimates of transmission value [p.3].
- For example, we find that the market value of transmission is highly influenced by the small fraction of time during which transmission is extremely valuable [p.15].
- Models could benchmark simulated transmission market values against empirical market outcomes both in terms of their average magnitude and distribution of values over time [p.15].

While this paper could have some relevance for individual investment decisions, we do not propose or assess any procedures for evaluating transmission investments.

The paper is very interesting, but it needs a tighter use of language to clarify the lessons learned.

We are glad the reviewer believes the paper is interesting and we have tightened the language used in response to the reviewer’s specific concerns, as described above.

Reviewer #2

We appreciate your efforts reviewing this work and the positive feedback!

Electric transmission value and its drivers in United States power markets

NCOMMS-24-16821B

Referee Report

5/25/25

Overview

This paper addresses the value of electric power transmission expansion based on an extensive analysis of historical locational prices in various Regional Transmission Organization (RTO) in the United States. The authors explain the difficulty of using prospective forecasts and cost-benefit analysis. The reliance on historical prices circumvents many of the problems in more conventional cost-benefit analyses of transmission expansion. The results compare different estimates of transmission value with external estimates of transmission expansion costs. The sensitivity analyses provide important conclusions such as the need to focus on real-time prices rather than day-ahead estimates of prices.

Details

There is a great deal of interest in the paper, and for most of the discussion the authors provide reasonable caveats to explain the limitations such as the difficulty of precisely estimating transmission expansion costs.

The basic methodology is to focus on locational prices, which include congestion costs, and to interpret the difference in prices between two locations as an estimate of the marginal value of transmission. This is true as a marginal value of energy transmission defined as injection at one location and withdrawal at another. However, this is not necessarily the same thing as the value of transmission line expansion.

The difficulty centers on two choices. First, “The transmission value analyzed in this section is a pairwise quantity between two wholesale market pricing nodes (i.e., a “link”) defined as the mean absolute locational marginal price difference between the nodes over time (in units \$/MWh).” This use of the term “link” could easily be misinterpreted to imply that there is a transmission line connecting the two locations and that the price difference between two locations would be caused by constraints on that line. But any “line may not be fully utilized even when there is congestion due to security constraints and power flow dynamics. The latter refers to the fact that the path of power flow on alternating current (AC) networks cannot be arbitrarily prescribed, because the electricity flows across transmission facilities according to Kirchoff’s Laws based on the power

consumption of loads and the voltage magnitude and real power injection established by generators.” In other words, the price difference may be, and typically is, caused by simultaneous limits on many transmission lines, including contingency security constraints.

Furthermore, transmission line expansion does not typically follow a constant-returns to scale cost structure. Hence the ex ante marginal conditions may not be an accurate indicator of beneficial transmission investment. In other words, material congestion costs are a necessary but not sufficient indicator of beneficial transmission investments.

This seems to take us back to the use of models comparing existing operations against a new profile created by a candidate transmission investment.

The paper is very interesting, but it needs a tighter use of language to clarify the lessons learned.